# Reaction Prediction via Interaction Modeling of Symmetric Difference Shingle Sets

**Runhan Shi[1], Letian Chen[1,2], Gufeng Yu[1], Yang Yang[1]\***
[1]AGI Institute, School of Computer Science, Shanghai Jiao Tong University
[2]Shanghai Innovation Institute
{han.run.jiangming, clt2001, jm5820zz}@sjtu.edu.cn
yangyang@cs.sjtu.edu.cn

## Abstract

Chemical reaction prediction remains a fundamental challenge in organic chemistry, where existing machine learning models face two critical limitations: sensitivity to input permutations (molecule/atom orderings) and inadequate modeling of substructural interactions governing reactivity. These shortcomings lead to inconsistent predictions and poor generalization to real-world scenarios. To address these challenges, we propose ReaDISH, a novel reaction prediction model that learns permutation-invariant representations while incorporating interaction-aware features. It introduces two innovations: (1) symmetric difference shingle encoding, which extends the differential reaction fingerprint (DRFP) by representing shingles as continuous high-dimensional embeddings, capturing structural changes while eliminating order sensitivity; and (2) geometry-structure interaction attention, a mechanism that models intra- and inter-molecular interactions at the shingle level. Extensive experiments demonstrate that ReaDISH improves reaction prediction performance across diverse benchmarks. It shows enhanced robustness with an average improvement of 8.76% on $R^2$ under permutation perturbations.[1]

## 1 Introduction

Accurate modeling of chemical reactions is a fundamental problem in organic chemistry, as it provides critical insights into reaction mechanisms, predicts reaction outcomes, and guides experimental design [1–4]. Reaction representation learning is central to tasks such as reaction yield prediction [5], enantioselectivity prediction [6], conversion rate estimation [7], and reaction type classification [8]. These tasks have gained considerable attention with the rise of machine learning (ML). Nevertheless, representing reactions for ML is challenging due to the complexity of reaction spaces and the multitude of factors influencing chemical experiments like substrates, catalysts, and reaction conditions [9–11]. While many reactions may appear theoretically feasible [12], in practice, successfully executing them requires a deeper understanding of how these factors interact. Even slight variations in any of these elements can significantly influence the outcome [13, 14], making the reaction prediction problem a complex and nuanced challenge.

Despite advances in ML models, they still face two major limitations that hinder their broader applicability, as shown in Figure 1. First, many models, especially those based on sequential representations like SMILES [15], fail to account for the inherent permutation invariance of chemical reactions [16, 17]. They tend to produce inconsistent predictions when changing the ordering of input molecules (e.g., swapping reactants and reagents) or the ordering of atoms (e.g., alternative

---

*Corresponding author. This work was supported by the National Natural Science Foundation of China (No. 62272300).

[1]The code is available at `https://github.com/Meteor-han/ReaDISH`.

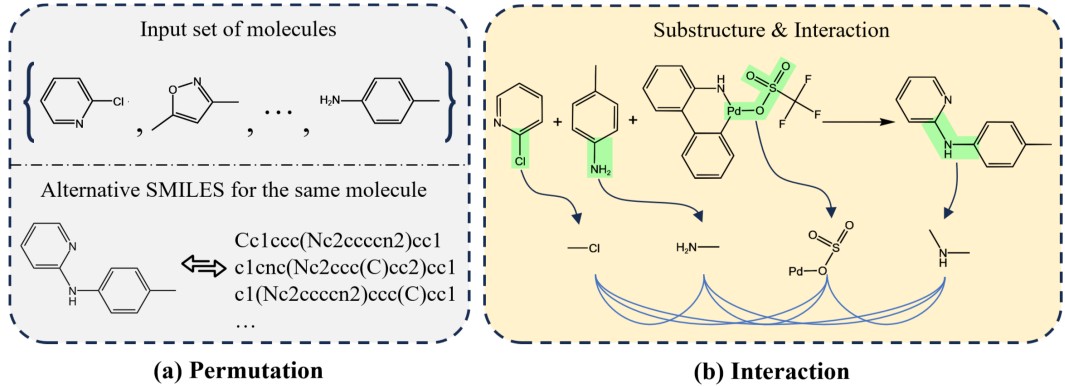

**(a) Permutation**    **(b) Interaction**

Figure 1: **Challenges for reaction representation learning**. (a) Permutation perturbations by inter-molecular order variation (top) and intra-molecular SMILES token randomization (bottom). (b) Key substructures that determine the outcomes of reactions and their inherent interactions.

SMILES). Such sensitivity to input ordering undermines reliability and generalizability. While data augmentation techniques can partially alleviate this issue, training on all possible permutations is computationally inefficient and impractical at scale [18, 19]. Achieving permutation invariance is essential for robust and unbiased reaction modeling across diverse chemical scenarios.

Second, current works often fail to capture the structural interactions that drive chemical reactivity in reaction modeling. Many approaches overlook critical substructures, particularly those that change from reactants to products (e.g., reaction centers [20]), and lack explicit modeling of interactions between such substructures. Although atom-level interactions have been widely explored in molecular representation learning [21–23], capturing higher-order interactions among chemically meaningful substructures across molecules remains limited in the reaction prediction problem. This lack of interaction awareness leads to suboptimal predictions and diminishes the model's ability to capture underlying reaction mechanisms [24]. As a result of these two limitations, many models perform poorly on out-of-sample reactions that better reflect real-world chemical diversity [25–27].

To address these challenges, we propose the ReaDISH model for **Rea**ction prediction via interaction modeling of symmetric **DI**fference **SH**ingle sets. First, we construct chemically meaningful substructure sets named *molecular shingle* [28–30] sets via the circular topology for each molecule to capture key structural components. Then we compute and encode the *symmetric difference* of shingle sets between reactants and products adopted from DRFP [31], which is naturally robustly permutation-invariant. Second, we add intra- and inter-molecular interaction pair representation on shingle-level attention based on geometric distance, structural distance, and chemical connectivity of shingles to improve interaction awareness. We conduct numerical experiments on various reaction representation learning tasks and benchmarks, including **reaction yield prediction**, **enantioselectivity prediction**, **conversion rate estimation**, and **reaction type classification**. Experimental results show that compared with baseline models, our proposed ReaDISH achieves SOTA on prediction accuracy and uncertainty estimation in most scenarios. It increases $R^2$ by an average of 8.76% under out-of-sample splits when performing permutation perturbations, presenting a better generalization capability. We present related work in Appendix A. Our key contributions are summarized as follows:

- We introduce a new way to present reaction structures based on the symmetric difference of molecular shingles between reactants and products, which is inherently permutation-invariant and highlights critical substructures that drive reaction outcomes.

- We design an interaction-aware attention mechanism that integrates geometric distance, structural distance, and chemical connectivity between shingles. This pair representation enables the model to capture both intra- and inter-molecular interactions.

- Extensive results across a wide range of tasks demonstrate that our proposed ReaDISH outperforms baseline models, achieving higher prediction accuracy and lower prediction uncertainty, especially under out-of-sample conditions.

## 2  Background

### 2.1  Problem definition

We denote the dataset of chemical reactions and their associated labels as $\mathcal{D} = \{(\mathcal{R}_i, y_i)\}_{i=1}^{N}$, where each reaction $\mathcal{R}_i$ consists of a set of participating molecules and $y_i \in \mathcal{Y}$ is the corresponding property label. Depending on the task, $y_i$ may represent (1) a real-valued scalar for regression tasks, such as reaction yield or energy barrier, i.e., $y_i \in \mathbb{R}$, or (2) a categorical label for classification tasks, such as reaction type or success/failure, i.e., $y_i \in \mathbb{N}$.

Formally, each reaction is denoted by $\mathcal{R}_i = \{\mathcal{M}_i^{\mathrm{r}}; \mathcal{M}_i^{\mathrm{p}}\} = \{M_i^{\mathrm{r}_1}, \ldots, M_i^{\mathrm{r}_m}; M_i^{\mathrm{p}_1}, \ldots, M_i^{\mathrm{p}_n}\}$, where $\mathcal{M}_i^{\mathrm{r}}$ ($\mathcal{M}_i^{\mathrm{p}}$) denotes the set of reactant (product) molecules and $m$ ($n$) is the number of reactant (product) molecules. For simplicity and consistency, we refer to molecules other than the products collectively as reactants (including catalysts, solvents, etc.). Each molecule $M_i^j$ is represented as the 3D conformer $C_i^j = \{(a_k, \mathbf{x}_k)\}_{k=1}^{N_{\mathrm{a}}}$, where $a_k$ is the atomic type, $\mathbf{x}_k \in \mathbb{R}^3$ is the spatial coordinate, and $N_{\mathrm{a}}$ is the number of atoms. The goal of reaction representation learning is to design a mapping function $f_\phi : \mathcal{R} \to \mathcal{Y}$, parameterized by $\phi$, such that the predicted label $\hat{y}_i = f_\phi(\mathcal{R}_i)$ approximates the true label $y_i$. During training, the model minimizes an objective function, such as root mean squared error (RMSE) for regression or cross-entropy loss for classification.

### 2.2  Molecular shingles

In cheminformatics, *molecular shingles* refer to structured fragments designed to capture local connectivity patterns in molecules. These shingles are typically defined as sets of atoms and their connectivity within a defined neighborhood, which are also utilized in classical fingerprints like ECFP [32], effectively representing the structural information of the molecule.

Formally, let a molecule $M$ be represented by a conformer $C = (V, E, \mathbf{X})$ with atoms $V$, bonds $E$, and coordinates $\mathbf{X}$. We define an $r$-sized shingle of an atom $v \in V$ as a connected subgraph induced by the center atom and its neighboring atoms with radius $r$, along with the corresponding bonds and their 3D positions. Let $\mathcal{N}_r(v)$ denote the set of neighboring atoms to atom $v$ with radius $r$. The corresponding shingle is then given by

$$S^{(r)}(v) = C\left[\{v\} \cup \mathcal{N}_r(v)\right], \tag{1}$$

where $C[\,\cdot\,]$ denotes the sub-conformer of $C$ restricted to the specified subset of atoms $\{v\} \cup \mathcal{N}_r(v)$. The set of all $r$-sized shingles in $M$ is denoted as $\mathcal{S}_M^{(r)}$, and its representation is invariant to the arrangement of atoms.

## 3  Method

### 3.1  Model architecture

We propose ReaDISH, a novel framework for reaction property prediction that learns permutation-invariant and interaction-aware representations of chemical reactions. The overall architecture is illustrated in Figure 2. It comprises three main components: an embedding layer that processes molecular inputs, an encoder that captures geometric and structural features, and a lightweight predictor that outputs reaction properties.

Given a reaction $\mathcal{R} = \{C_i^{\mathrm{r}_1}, \ldots, C_i^{\mathrm{r}_m}; C_i^{\mathrm{p}_1}, \ldots, C_i^{\mathrm{p}_n}\}$ consisting of reactant and product molecules in 3D conformer format, ReaDISH first encodes each molecule $C_i^j$ using a 3D molecular encoder $f_{\mathrm{mol}}$ to generate atom-level representations $\mathbf{X}^{\mathrm{a}} \in \mathbb{R}^{N_{\mathrm{a}} \times F}$, where $F$ is the embedding dimension. A shingle-generation algorithm then extracts molecular shingles and computes the *symmetric difference* between reactant and product shingle sets to capture reaction-specific transformations. Next, ReaDISH aggregates the atom-level representations within each shingle through a pooling operation, producing initial shingle-level embeddings $\mathbf{X}^0 \in \mathbb{R}^{N_{\mathrm{s}} \times F}$, where $N_{\mathrm{s}}$ is the number of shingles. We compute three types of pairwise relations to model interactions between shingles: geometric distances, structural distances, and chemical connectivity. These are encoded as pairwise matrices $\mathbf{P} = \{\mathbf{P}_{\mathrm{g}}, \mathbf{P}_{\mathrm{e}}, \mathbf{P}_{\mathrm{s}}\} \in \mathbb{R}^{3 \times N_{\mathrm{s}} \times N_{\mathrm{s}}}$. To incorporate these relationships into the attention mechanism, we apply $K$ Gaussian kernels to each pairwise matrix to produce the initial pair representation $\mathbf{P}^0$. The ReaDISH encoder

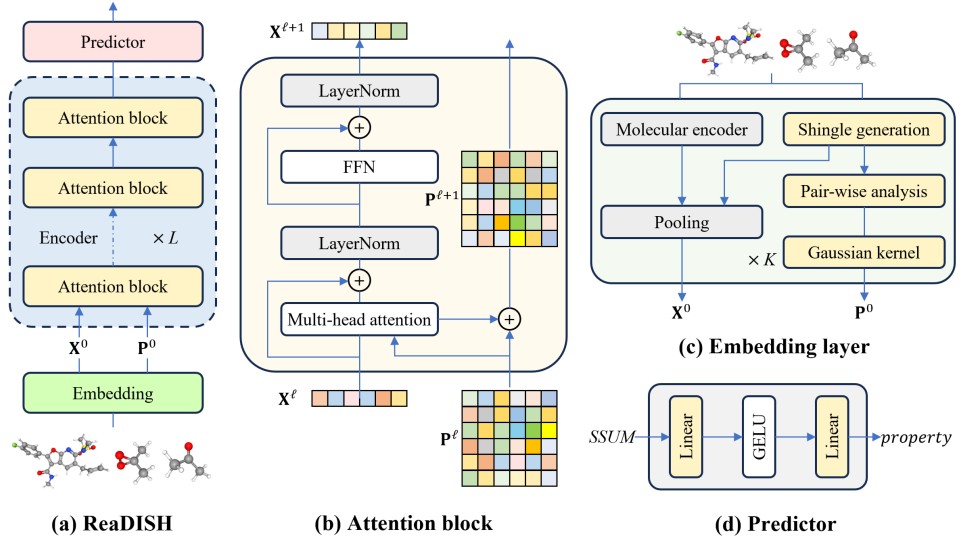

**(a) ReaDISH**    **(b) Attention block**    **(d) Predictor**

**(c) Embedding layer**

Figure 2: **Overall architecture of ReaDISH** (a). It consists of an embedding layer (c), an encoder incorporating $L$ attention blocks (b), where the extended self-attention module with the Gaussian kernel is depicted in Figure 4(a), and a lightweight predictor (d) for predicting reaction properties.

then processes the shingle-level representations using $L$ transformer-style [33] attention blocks. Each block contains a multi-head attention (MHA) layer that integrates the pairwise attention biases $\mathbf{P}^\ell$ at each layer $\ell$. Finally, ReaDISH uses a special `[SSUM]` token to summarize the reaction representation and passes it to a multilayer perceptron (MLP) head to predict reaction property $\hat{y}$.

## 3.2 Embedding layer for shingles

**Molecular encoder.** The 3D molecular encoder $f_{\mathrm{mol}}$ transforms each molecule $C = \left\{(a_k, \mathbf{x}_k)\right\}_{k=1}^{N_a}$, comprising atom types $a_k$ and 3D coordinates $\mathbf{x}_k$, into atom-level feature representations $\mathbf{X}^a \in \mathbb{R}^{N_a \times F}$ followed by layer normalization:

$$\mathbf{X}^a = \mathrm{LayerNorm}\left(f_{\mathrm{mol}}(C)\right). \tag{2}$$

**Symmetric difference shingle set generation.** In reaction modeling, comparing shingles across reactants and products allows for the precise identification of structural transformations [31]. For each reaction, we treat molecules except products as reactants. Shingles within a given radius are generated for reactants and products, respectively. The generated shingles are expressed in SMILES strings to perform the symmetric difference operation, as shown in Figure 3. This operation yields the set of shingles that are exclusive to either reactants or products, thereby identifying structural changes associated with the reaction mechanism. The complete algorithm and illustration for computing the symmetric difference shingle set are provided in Appendix C. Formally, for a reaction $\mathcal{R} = \{\mathcal{M}^r; \mathcal{M}^p\}$ and a radius $r_{\max}$, we define the symmetric difference shingle set $\mathcal{S}_\mathcal{R}$ as

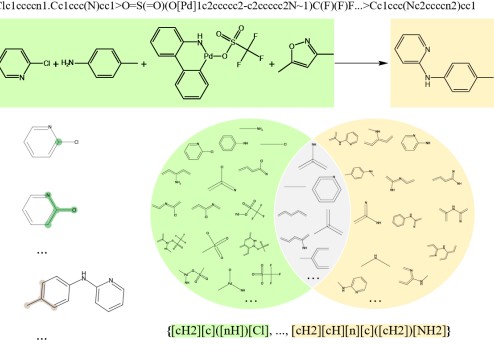

Figure 3: **Shingles generation**. We remove the intersection part (in gray), and keep the remaining shingles for reactants (in green) and products (in yellow).

$$\mathcal{S}_\mathcal{R} = \mathcal{S}_{\mathrm{react}} \bigtriangleup \mathcal{S}_{\mathrm{prod}} = \left(\bigcup_{i=1}^{r_{\max}} \bigcup_{j=1}^{m} \mathcal{S}_{M^{r_j}}^{(i)}\right) \bigtriangleup \left(\bigcup_{i=1}^{r_{\max}} \bigcup_{k=1}^{n} \mathcal{S}_{M^{p_k}}^{(i)}\right), \tag{3}$$

where $\mathcal{S}_{\text{react}}$ ($\mathcal{S}_{\text{prod}}$) is the shingle set for reactants (products), and $\triangle$ denotes the symmetric difference operator $\mathcal{A} \triangle \mathcal{B} = (\mathcal{A} \setminus \mathcal{B}) \cup (\mathcal{B} \setminus \mathcal{A})$. This set-based strategy is invariant to the ordering of molecules.

**Shingle pooling.** To construct shingle-level representations from atom-level embeddings, we apply a shingle pooling operation over the set of symmetric difference shingles. Each shingle $S \in \mathcal{S}_{\mathcal{R}}$ consists of a subset of atoms, and its representation is computed by averaging the embeddings:

$$\mathbf{h}_S = \text{Pooling}\,(S, \mathbf{X}^{\text{a}}) = \frac{1}{|\,\mathcal{A}(S)\,|} \sum_{i \in \mathcal{A}(S)} \mathbf{x}_i^{\text{a}}, \tag{4}$$

where $\mathcal{A}(S) \subseteq \{1, \dots, N_{\text{a}}\}$ denotes the indices of atoms in shingle $S$ and $\mathbf{x}_i^{\text{a}}$ is the embedding of the $i$-th atom. This operation aggregates localized chemical information within each shingle, enabling the model to learn chemically meaningful shingle-level representations. The resulting set of initial shingle-level representations is denoted as

$$\mathbf{X}^0 = [\,\mathbf{h}_{S_i}\,]_{i=1}^{N_{\text{s}}} \in \mathbb{R}^{N_{\text{s}} \times F}. \tag{5}$$

These pooled embeddings serve as the input to the subsequent encoder, which models interactions between shingles to capture the overall reaction context.

**Pairwise interaction.** We introduce an interaction framework that explicitly captures pairwise dependencies between molecular shingles. This formulation allows ReaDISH to model both intra- and inter-molecular interactions, leveraging geometric and structural features to represent chemical transformations. As illustrated in the right panel of Figure 4(b), we define three pairwise metrics between shingles $S_i$ and $S_j$ as

$$d_{ij}^{\text{g}} = \begin{cases} \|\mathbf{c}_i - \mathbf{c}_j\|_2, & \text{if } S_i, S_j \text{ belong to the same molecule,} \\ 0, & \text{otherwise;} \end{cases}$$

$$d_{ij}^{\text{e}} = \begin{cases} 1, & \text{if } S_i, S_j \text{ belong to the same molecule,} \\ 0, & \text{otherwise;} \end{cases} \tag{6}$$

$$d_{ij}^{\text{s}} = 1 - \text{sim}(S_i, S_j),$$

where $\mathbf{c}_i$ and $\mathbf{c}_j$ denote the geometric centers of shingles $S_i$ and $S_j$, respectively, and $\text{sim}(\cdot, \cdot)$ is the Tanimoto similarity [34] of their Morgan fingerprints [35]. These metrics yield a structured pairwise representation $\mathbf{P} = \{\mathbf{P}_{\text{g}}, \mathbf{P}_{\text{e}}, \mathbf{P}_{\text{s}}\} = \left[\,d_{ij}^{\text{g}}, d_{ij}^{\text{e}}, d_{ij}^{\text{s}}\,\right]_{i,j=1}^{N_{\text{s}}} \in \mathbb{R}^{3 \times N_{\text{s}} \times N_{\text{s}}}$, where each component captures a specific interaction modality across all $N$ shingles. The geometric distance $d_{ij}^{\text{g}}$ encodes spatial relationships within the same molecule, analogous to atom-level distance modeling. The binary chemical connectivity $d_{ij}^{\text{e}}$ indicates whether two shingles are part of the same molecule. The structural distance $d_{ij}^{\text{s}}$ emphasizes functional dissimilarity, which is essential for modeling reactive interactions across different molecules. These pair interactions enable a finer-grained understanding of reaction mechanisms and facilitate more accurate prediction of reaction outcomes.

**Gaussian kernel.** After computing the pairwise interactions, we integrate these geometry-aware and structure-aware signals into the attention mechanism to better capture chemically meaningful context. To this end, we adopt the Gaussian Kernel with Pair Type (GKPT) [36], a technique effectively applied in molecular representation learning [21]. GKPT applies an affine transformation to pairwise distances based on the interaction type, followed by a classical Gaussian kernel, as illustrated in Figure 4(a). Formally, the GKPT is defined as

$$\text{GKPT}\,((x, e), \boldsymbol{\mu}, \boldsymbol{\sigma}) = \mathcal{G}\,(\mathbf{E}_1(e) \cdot x + \mathbf{E}_2(e), \boldsymbol{\mu}, \boldsymbol{\sigma})\,, \tag{7}$$

where $\mathcal{G}\,(x', \boldsymbol{\mu}, \boldsymbol{\sigma}) = \frac{1}{\sqrt{2\pi}\boldsymbol{\sigma}} \exp\left(-\frac{1}{2}\left(\frac{x'-\boldsymbol{\mu}}{\boldsymbol{\sigma}}\right)^2\right)$ is the standard Gaussian kernel. Here, $x$ is the input distance, $e$ is the pair type index, $\mathbf{E}_1, \mathbf{E}_2 \in \mathbb{R}^{N_{\text{e}} \times K}$ are learnable embedding layers, $N_{\text{e}}$ is the number of pair types, and $\boldsymbol{\mu}, \boldsymbol{\sigma} \in \mathbb{R}^K$ are learnable parameters for $K$ Gaussian kernels. We apply separate GKPT modules to process geometric distance ($d_{ij}^{\text{g}}$) and structural distance ($d_{ij}^{\text{s}}$) along with chemical connectivity ($d_{ij}^{\text{e}}$) as shown in Figure 4(b). Each output is projected into the attention space and combined to form the pairwise bias term as

$$p_{ij} = \text{GKPT}_{\text{g}}\,\left((d_{ij}^{\text{g}}, d_{ij}^{\text{e}}), \boldsymbol{\mu}, \boldsymbol{\sigma}\right) \mathbf{w}_{\text{g}} + \text{GKPT}_{\text{s}}\,\left((d_{ij}^{\text{s}}, d_{ij}^{\text{e}}), \boldsymbol{\mu}, \boldsymbol{\sigma}\right) \mathbf{w}_{\text{s}}, \tag{8}$$

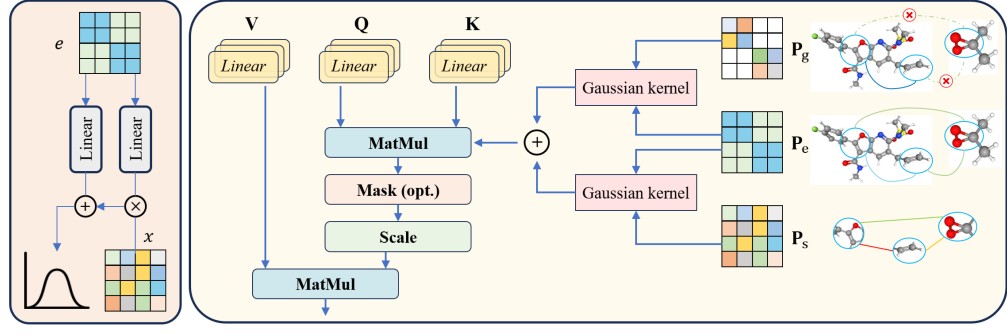

**(a) Gaussian kernel**   **(b) Geometry- and structure-enhanced attention**

Figure 4: **Interaction-aware attention**. (a) Gaussian kernel with learned pair type transformations. (b) Self-attention enhanced by geometric and structural interactions. Each pairwise representation incorporates one intra-molecular relationship (geometric distance) and two inter-molecular relationships (structural distance and chemical connectivity).

where $\mathbf{w}_g, \mathbf{w}_s \in \mathbb{R}^K$ are learnable projection vectors. The full interaction-aware pairwise bias matrix used in attention is denoted as

$$\mathbf{P}^0 = \left[\, p_{ij} \,\right]_{i,j=1}^{N_s} \in \mathbb{R}^{N_s \times N_s}. \tag{9}$$

This pairwise representation encodes chemically relevant interactions and informs the subsequent attention computation, enabling the model to learn complex spatial and structural dependencies in chemical reactions.

### 3.3 Enhanced attention block

To incorporate geometric and structural information, we introduce an attention mechanism enhanced by pair representation. Specifically, we augment the standard self-attention with a learnable pairwise bias derived from the previously defined pairwise representations. At each layer $\ell$, we update the pairwise bias $p_{ij}^\ell$ between shingle $S_i$ and $S_j$ using the query-key interaction:

$$p_{ij}^{\ell+1} = \frac{\mathbf{Q}_i^\ell \left(\mathbf{K}_j^\ell\right)^\top}{\sqrt{d}} + p_{ij}^\ell, \tag{10}$$

where $\mathbf{Q}_i^\ell$ and $\mathbf{K}_j^\ell$ are the query and key vectors of shingles $i$ and $j$ at layer $\ell$, respectively, and $d$ is the hidden dimension. The attention score between shingles $i$ and $j$ is then computed by incorporating this updated bias into the scaled dot-product attention:

$$\text{Attention}\left(\mathbf{Q}_i^\ell, \mathbf{K}_j^\ell, \mathbf{V}_j^\ell\right) = \text{softmax}\left(\frac{\mathbf{Q}_i^\ell \left(\mathbf{K}_j^\ell\right)^\top}{\sqrt{d}} + p_{ij}^{\ell-1}\right)\mathbf{V}_j^\ell, \tag{11}$$

where $\mathbf{V}_j^\ell$ is the value vector of shingle $j$. This formulation allows the attention mechanism to be guided by chemically meaningful geometric and structural priors. Additionally, we prepend a learnable [SSUM] token to the shingle sequence $\mathbf{X}^\ell$ at each layer, which serves as a global summary token for downstream reaction property prediction.

### 3.4 Predictor

ReaDISH utilizes a lightweight predictor, as depicted in Figure 2(d), to predict reaction properties. It consists of two linear layers with GELU activation [37]. This simple yet effective architecture maps the final [SSUM] token representation, which encodes the global context of the reaction, to the target prediction space.

### 3.5 Pre-training via pseudo-reaction-type classification

To enhance the generalization capability of ReaDISH, we introduce a pseudo-reaction-type classification task as the pre-training objective on a large-scale dataset. Specifically, we assign each reaction

a pseudo-label by clustering its structural representation at different granularities. This multi-scale classification setting encourages the encoder to capture both coarse-grained and fine-grained structural semantics. We generate $K_t$ pseudo-labels for each reaction. Formally, the pre-training loss is defined as

$$\mathcal{L}_{\text{pseudo}} = \frac{1}{N} \sum_{n=1}^{N} \sum_{k=1}^{K_t} \mathcal{L}\left(\mathbf{W}_k(f_\phi(\mathbf{X}_n^0, \mathbf{P}_n^0)), y_n^{(k)}\right), \tag{12}$$

where $f_\phi$ denotes the ReaDISH encoder, $\mathcal{L}$ is the cross-entropy loss, $\mathbf{W}_k \in \mathbb{R}^{F \times N_k}$ with target dimension $N_k$ for $k \in \{1, \ldots, K_t\}$ are parameters of fully connected layers, respectively, and $y_n^{(k)} \in \mathbb{N}$ are pseudo-labels for the $n$-th sample. More information can be found in Appendix D.

## 4 Experiments

### 4.1 Settings

**Pre-training datasets.** We collect 3.7M chemical reactions for pre-training based on the United States Patent and Trademark Office (USPTO) dataset [38] and the Chemical Journals with High Impact Factor (CJHIF) dataset [39]. We employ the DRFP [31] method with the $K$-means algorithm to compute pseudo-labels. More information can be found in Appendix B.

**Downstream datasets.** To comprehensively evaluate the effectiveness of ReaDISH, we use seven datasets across a wide range of chemical tasks, including: (1) yield prediction, the Buchwald-Hartwig (BH) dataset [13], the Suzuki-Miyaura (SM) dataset [14], the real-world electronic laboratory notebook (ELN) dataset [40], and the Ni-catalyzed C-O bond activation (NiCOlit) dataset [41]; (2) enantioselectivitiy prediction, the asymmetric N,S-acetal formation (N,S-acetal) dataset [42]; (3) conversion rate estimation, the C-heteroatom-coupling reactions (C-heteroatom) dataset [43]; and (4) reaction type classification, the USPTO_TPL dataset [8]. We consider both random and more challenging out-of-sample splits, which better reflect practical deployment scenarios. More information can be found in Appendix B.

**Baselines and metrics.** We use five representative models as baselines in our experiments: Yield-BERT [5], YieldBERT-DA [44], UA-GNN [45], UAM [46], and ReaMVP [47], which we introduce in related work in Appendix A. We additionally compare our method with a cross-attention baseline that directly models reactant-product interactions as provided in Appendix E. Given the importance of spatial structural patterns in molecules, we adopt the SE(3)-invariant [48] Uni-Mol [21] as our 3D molecular encoder $f_{\text{mol}}$ for its excellent performance. For regression tasks, we report mean absolute error (MAE $\downarrow$), root mean squared error (RMSE $\downarrow$), and coefficient of determination (R$^2$ $\uparrow$). For classification tasks, we measure accuracy (ACC $\uparrow$), Matthews correlation coefficient (MCC $\uparrow$), and confusion entropy (CEN $\downarrow$). More information can be found in Appendix F.

### 4.2 Main results

**ReaDISH achieves SOTA or competitive performance across diverse benchmark datasets.** As shown in Figure 5(a), ReaDISH achieves top or competitive results on six datasets under random splits. It ranks first on the BH, N,S-acetal, and C-heteroatom datasets for R$^2$, and the USPTO_TPL dataset for accuracy. On the SM and NiCOlit datasets, ReaDISH consistently matches or outperforms existing models. Figure 5(b) presents results on more challenging out-of-sample splits, where the test distribution differs from the training one. ReaDISH consistently outperforms all baseline models across six benchmark datasets. It shows a greater advantage under out-of-sample splits, which better reflects real-world scenarios and offers greater practical relevance than random splits, demonstrating ReaDISH's strong robustness across diverse reaction types and chemical spaces.

On the particularly challenging ELN dataset, however, all methods show limited performance with R$^2$ below 0.3. This underscores the difficulty in generalizing to complex, real-world reaction data. The poor results can be attributed to two main factors: (1) the ELN dataset includes a broader range of substrates, ligands, and solvents than other datasets, with significantly greater structural variability, making it harder for models to capture meaningful patterns [26]; and (2) limited fine-tuning data, which constrains the model's ability to adapt to the diverse chemical contexts. Performance regarding training size is presented in Appendix G. Full results are available in Appendix H.

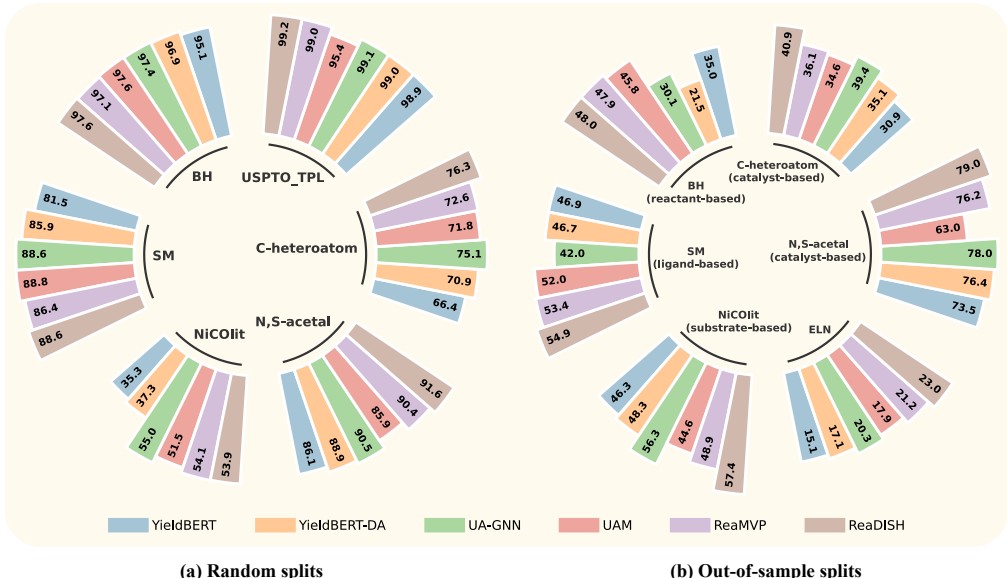

(a) Random splits          (b) Out-of-sample splits

Figure 5: **Performance comparison**. (a) Results under random splits for six datasets and (b) results under out-of-sample splits for six datasets, where we report accuracy (%) for the USPTO_TPL dataset and $R^2$ (%) for other datasets.

Table 1: Impact of permutation-invariant modeling on prediction uncertainty scores in out-of-sample splits. [a] denotes methods that perform permutation data augmentation, and [b] denotes permutation-invariant methods. Bold entries highlight the best performance.

| Method | BH (reactant-based) | | | SM (ligand-based) | | |
|---|---|---|---|---|---|---|
| | MAE | RMSE | $R^2$ | MAE | RMSE | $R^2$ |
| YieldBERT | $17.65 \pm 1.13$ | $23.96 \pm 1.17$ | $0.342 \pm 0.082$ | $14.98 \pm 0.38$ | $20.05 \pm 0.39$ | $0.447 \pm 0.029$ |
| YieldBERT-DA[a] | $18.41 \pm 0.48$ | $26.30 \pm 0.33$ | $0.215 \pm 0.020$ | $15.78 \pm 0.24$ | $19.64 \pm 0.26$ | $0.467 \pm 0.014$ |
| UA-GNN[b] | $16.95 \pm 0.21$ | $24.82 \pm 0.63$ | $0.301 \pm 0.035$ | $15.59 \pm 0.36$ | $20.49 \pm 0.39$ | $0.420 \pm 0.022$ |
| UAM | $17.34 \pm 0.63$ | $22.61 \pm 1.24$ | $0.421 \pm 0.072$ | $15.84 \pm 0.53$ | $19.61 \pm 0.53$ | $0.503 \pm 0.042$ |
| ReaMVP | $17.62 \pm 1.04$ | $22.52 \pm 1.47$ | $0.432 \pm 0.068$ | $13.91 \pm 0.29$ | $18.64 \pm 0.40$ | $0.516 \pm 0.026$ |
| ReaDISH[b] | $\mathbf{16.29 \pm 0.30}$ | $\mathbf{21.76 \pm 0.80}$ | $\mathbf{0.480 \pm 0.032}$ | $\mathbf{14.22 \pm 0.33}$ | $\mathbf{18.05 \pm 0.37}$ | $\mathbf{0.549 \pm 0.019}$ |

**Permutation-invariant modeling enhances prediction robustness.** A key strength of ReaDISH is its permutation-invariant design, which ensures consistent predictions regardless of the ordering of input molecules or SMILES tokens. To test whether this property enhances robustness in out-of-sample settings, we measure the standard deviation of model predictions across five runs, each using different random molecule orderings and SMILES permutations, the same as YieldBERT-DA [44]. As shown in Table 1, ReaDISH consistently shows the lowest prediction variance. It increases $R^2$ by 11.11% and 6.40% on the BH and SM out-of-sample splits (average 8.76%), respectively. In contrast, models without permutation invariance show greater variability and reduced performance under input perturbations. We show an example in Figure 6.

These findings highlight the practical benefits of permutation invariance in enhancing model reliability. By modeling interactions between symmetric difference shingles in an order-independent manner, ReaDISH offers more stable and reliable predictions under out-of-sample conditions.

## 4.3 Ablation study

To evaluate the contributions of individual components within ReaDISH, we conduct a series of ablation studies concerning pair representation, symmetric difference, pre-training strategies, and radius of shingles. Table 2 presents the results for the BH and SM datasets under out-of-sample splits.

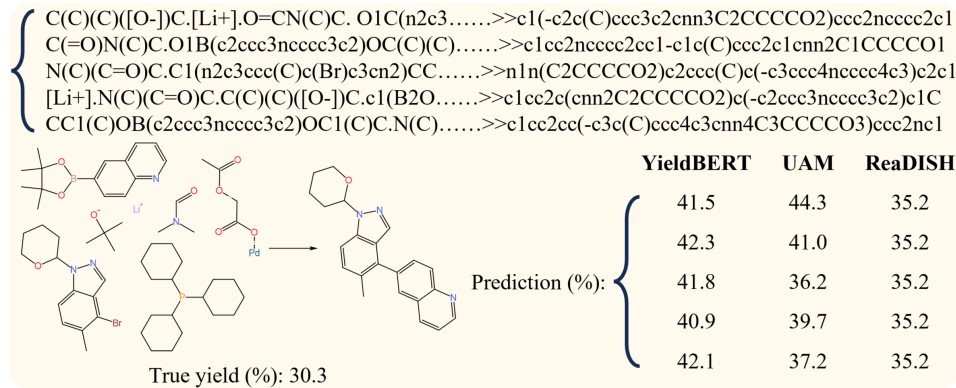

$$\begin{cases} \text{C(C)(C)([O-])C.[Li+].O=CN(C)C. O1C(n2c3......>>c1(-c2c(C)ccc3c2cnn3C2CCCCO2)ccc2ncccc2c1} \\ \text{C(=O)N(C)C.O1B(c2ccc3ncccc3c2)OC(C)(C)......>>c1cc2ncccc2cc1-c1c(C)ccc2c1cnn2C1CCCCO1} \\ \text{N(C)(C=O)C.C1(n2c3ccc(C)c(Br)c3cn2)CC......>>n1n(C2CCCCO2)c2ccc(C)c(-c3ccc4ncccc4c3)c2c1} \\ \text{[Li+].N(C)(C=O)C.C(C)(C)([O-])C.c1(B2O......>>c1cc2c(cnn2C2CCCCO2)c(-c2ccc3ncccc3c2)c1C} \\ \text{CC1(C)OB(c2ccc3ncccc3c2)OC1(C)C.N(C)......>>c1cc2cc(-c3c(C)ccc4c3cnn4C3CCCCO3)ccc2nc1} \end{cases}$$

| | **YieldBERT** | **UAM** | **ReaDISH** |
|---|---|---|---|
| | 41.5 | 44.3 | 35.2 |
| | 42.3 | 41.0 | 35.2 |
| Prediction (%): | 41.8 | 36.2 | 35.2 |
| | 40.9 | 39.7 | 35.2 |
| | 42.1 | 37.2 | 35.2 |

True yield (%): 30.3

Figure 6: **Permutation influence on predictions**. We sample one reaction from the SM dataset under the out-of-sample split and perform five permutations on the molecule and SMILES token orderings with consistent conformers. While YieldBERT and UAM produce varying predictions with standard deviations of 0.49 and 2.88, respectively, ReaDISH consistently yields an invariant result with the least error.

Table 2: Ablation study results on key components of ReaDISH, evaluated under out-of-sample splits for the BH and SM datasets. Bold entries highlight the best performance.

| Method | BH (reactant-based split) | | | SM (ligand-based split) | | |
|---|---|---|---|---|---|---|
| | MAE | RMSE | $R^2$ | MAE | RMSE | $R^2$ |
| w/o $P^0$ | $16.82 \pm 0.34$ | $22.24 \pm 0.87$ | $0.430 \pm 0.038$ | $14.81 \pm 0.29$ | $19.24 \pm 0.40$ | $0.506 \pm 0.026$ |
| w/o $P_g$ | $16.55 \pm 0.44$ | $22.10 \pm 0.91$ | $0.457 \pm 0.040$ | $14.47 \pm 0.40$ | $18.36 \pm 0.42$ | $0.535 \pm 0.028$ |
| w/o $P_s$ | $16.41 \pm 0.36$ | $21.87 \pm 0.78$ | $0.464 \pm 0.034$ | $14.30 \pm 0.31$ | $18.30 \pm 0.41$ | $0.530 \pm 0.023$ |
| w/o $\triangle$ | $16.52 \pm 0.45$ | $21.95 \pm 0.98$ | $0.465 \pm 0.039$ | $14.35 \pm 0.37$ | $18.21 \pm 0.41$ | $0.538 \pm 0.017$ |
| w/o pre-training | $16.40 \pm 0.35$ | $21.80 \pm 0.85$ | $0.472 \pm 0.035$ | $14.26 \pm 0.32$ | $18.10 \pm 0.43$ | $0.540 \pm 0.020$ |
| ReaDISH ($r$=2) | $16.50 \pm 0.15$ | $22.11 \pm 0.25$ | $0.465 \pm 0.014$ | $14.41 \pm 0.33$ | $18.57 \pm 0.44$ | $0.523 \pm 0.228$ |
| ReaDISH ($r$=4) | $17.91 \pm 0.40$ | $23.90 \pm 0.01$ | $0.421 \pm 0.006$ | $14.88 \pm 0.56$ | $18.89 \pm 0.82$ | $0.505 \pm 0.043$ |
| ReaDISH ($r$=3) | $\mathbf{16.29 \pm 0.30}$ | $\mathbf{21.76 \pm 0.80}$ | $\mathbf{0.480 \pm 0.032}$ | $\mathbf{14.22 \pm 0.33}$ | $\mathbf{18.05 \pm 0.37}$ | $\mathbf{0.549 \pm 0.019}$ |

**(i) Impact of pair representation.** A central feature of ReaDISH is its use of pair representation to capture interactions between molecular shingles. We evaluate three ablated variants: (1) w/o $P^0$, which entirely removes pairwise modeling from self-attention, (2) w/o $P_g$, which excludes geometric distance information, and (3) w/o $P_s$, which omits structural distance features. The results show that removing the full pair representation causes a clear performance drop. Excluding either geometric or structural features also leads to consistent degradation in performance.

**(ii) Efficacy of symmetric difference shingles.** To test the effectiveness of the symmetric difference encoding strategy, we compare against a variant (w/o $\triangle$) that processes all shingles from both reactants and products without filtering duplicates. This modification leads to a noticeable increase in error, suggesting that without explicitly focusing on molecular transformations, the model is exposed to noise from irrelevant or redundant molecular features, which hampers learning.

**(iii) Contribution of pre-training.** We evaluate a version trained from scratch (w/o pre-training) to measure the effectiveness of the pseudo-reaction-type pre-training task. Removing this pre-training leads to reduced performance across the board. Though the performance gap is modest compared to other ablations, it highlights that even a simple pre-training task helps the model converge to better representations of the reaction space.

**(iv) Impact of shingle radius.** We assess how varying the maximum shingle generation radius affects downstream performance. Using a radius of 4 yields the poorest performance, likely because it produces excessive shingles, introducing redundancy and higher complexity. A radius of 2 performs slightly worse than radius 3, indicating that radius 3 offers a good balance between coverage and efficiency.

# 5 Conclusion

In this work, we present ReaDISH, a novel method for chemical reaction prediction that leverages the symmetric difference of molecular shingles to model chemical interactions. Our method incorporates geometric and structural information through a shingle-level attention mechanism that captures both intra- and inter-molecular interactions. Through comprehensive experiments across multiple reaction prediction tasks, we demonstrate that ReaDISH consistently outperforms baseline models and exhibits improved robustness to input permutations, particularly under out-of-sample scenarios. ReaDISH offers a flexible and effective framework for reaction modeling and paves the way for more interpretable and generalizable machine learning models in computational chemistry. We discuss limitations and impact statements in Appendix I and J, respectively. Future directions include extending ReaDISH to more complex tasks, such as retrosynthesis planning and the incorporation of reaction conditions to enable more accurate predictions.

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

# Appendix

## A  Related work

**Reaction performance prediction.**  Chemical reactions involve complex transformations among multiple molecular entities, such as reactants, catalysts, and reagents. Accurate prediction of reaction outcomes requires expressive and chemically informed representations. Early approaches rely heavily on handcrafted features as molecular fingerprint features derived from domain knowledge, to encode atomic and physicochemical properties [13, 14, 49, 40, 41]. Recent advances in deep learning for reaction prediction can be grouped into three main categories: sequence-based, graph-based, and conformer-based models. Sequence-based models represent chemical reactions using linear notations such as SMILES and apply neural sequence architectures to learn patterns in tokenized strings. Notable examples include YieldBERT [5] and YieldBERT-DA [44], which leverage BERT-based [50] encoders; MolFormer [19], which adopts a transformer [33] architecture; and T5Chem [51], which builds on the T5 language model [52]. Graph-based methods encode molecular structures as graphs and extract meaningful representations using graph neural networks (GNNs). Rxn Hypergraph [17] employs hypergraph attention to capture molecular interactions. UA-GNN [45] sums reactant embeddings via GNNs and concatenates them with product representations. UAM [46] combines multi-view inputs, including graphs, SMILES, and molecular fingerprints, and aggregates their embeddings. LocalTransform [53] focuses on local structural changes by encoding atom-level differences between reactants and products. Conformer-based models incorporate 3D atomic coordinates to learn geometry-aware representations, which can better capture stereoelectronic factors influencing reactivity. ReaMVP [47] combines Bi-GRU networks [54] with multi-view pre-training on conformers and SMILES. YieldFCP [55] introduces a cross-modal projector to align conformer and SMILES representations for yield prediction. Despite their respective advantages, these methods often overlook the rich intra- and inter-molecular interactions that influence reaction outcomes. In this work, we address this gap by introducing a geometry- and structure-enhanced modeling approach that explicitly incorporates such interactions into the representation learning process.

**Molecular substructure learning.**  Substructures (functional groups, motifs, and fragments) play a pivotal role in determining molecular properties and, by extension, reaction outcomes. Several fragment-based approaches have been proposed to decompose molecules into chemically meaningful parts. RECAP [56] introduces predefined cleavage rules to generate fragments at drug-like bond types, while BRICS [57] focuses on retrosynthetically relevant splits. ReLMole [58] further generalizes this process using graph-based heuristics to automatically extract relevant substructures. In parallel, fingerprinting methods such as ECFP [32] iteratively encode local atomic environments, capturing circular subgraphs of increasing radii. These subgraphs, referred to as molecular shingles [28–30], serve as a compact and effective representation of chemical structures. DRFP [31] extracts shingles and applies a symmetric difference operation on reactant and product shingle sets to obtain reaction fingerprints. Unlike DRFP, our method directly models the interaction-aware symmetric difference via a transformer, allowing it to capture fine-grained transformations at the shingle level.

## B  Datasets statistics

We use two datasets for pre-training and seven datasets for downstream evaluation, as summarized in Table 3. We remove duplicate records and invalid reactions for pre-training by RDKit [59]. To assess the generalizability of our approach, we consider both random and out-of-sample splits. In the out-of-sample split, the test set contains reactions involving molecules that do not appear in the training set. This setup mimics real-world scenarios, where unseen molecular structures are encountered during reaction property prediction. To strike a balance between accuracy in spatial coordinates and computational efficiency, we utilize the ETKDG algorithm [60] to generate up to 100 conformers and sample one for each molecule.

- **The United States Patent and Trademark Office (USPTO) dataset** [38] was collected from 1976 to September 2016, containing over 1.8 million chemical reactions stored in the form of SMILES arbitrary target specification (SMARTS).

Table 3: The statistics of pre-training datasets (first row) and evaluation datasets (remaining rows).

| Dataset | No. reactions | Split type | Out-of-sample | No. training | No. test |
|---|---|---|---|---|---|
| USPTO [38] & CJHIF [39] | 3,728,503 | stratified | ✗ | 3,542,077 | 186,426 |
| BH [13] | 3,955 | random | ✗ | 2,768 | 1,187 |
| | 3,955 | reactant-based | ✓ | 2,372 | 1,583 |
| SM [14] | 5,760 | random | ✗ | 4,032 | 1,728 |
| | 5,760 | ligand-based | ✓ | 4,320 | 1,440 |
| NiCOlit [41] | 1,406 | random | ✗ | 1,124 | 282 |
| | 1,406 | substrate-based | ✓ | 1,012 | 394 |
| ELN [40] | 750 | random | ✓ | 525 | 225 |
| N,S-acetal [42] | 1,075 | random | ✗ | 600 | 475 |
| | 688 | catalyst-based | ✓ | 384 | 304 |
| C-heteroatom [43] | 1,536 | random | ✗ | 1,075 | 461 |
| | 1,536 | catalyst-based | ✓ | 1,152 | 384 |
| USPTO_TPL [8] | 445,115 | random | ✗ | 400,604 | 44,511 |

- **The Chemical Journals with High Impact Factor (CJHIF) dataset** [39] included over 3.2 million chemical reactions in the form of SMARTS extracted from chemistry journals by Chemical.AI.

- **The Buchwald-Hartwig (BH) dataset** [13] contained 3,955 reactions from high-throughput experiments (HTEs) with 1,536-well plates on the class of Pd-catalyzed Buchwald-Hartwig C-N cross-coupling reactions. We choose the pyridyl reactants as the pivot to construct the out-of-sample split condition, where training includes nine pyridyl reactants and testing uses three non-pyridyl reactants, as shown in Figure 8(a).

- **The Suzuki-Miyaura (SM) dataset** [14] was constructed from high-throughput experiments on the class of Suzuki-Miyaura cross-coupling reactions, resulting in measured yields for a total of 5,760 reactions. We choose a set of ligands as the pivot to construct the out-of-sample split condition, where nine ligands (including "None") are used for training and three for testing, as shown in Figure 8(b).

- **The Ni-catalyzed C-O bond activation (NiCOlit) dataset** [41] was extracted from organic reaction publications to form C-C and C-N bonds, containing 1,406 reactions. We choose the OPiv substrates as the pivot to construct the out-of-sample split condition, where 247 OPiv substrates are used for training and 42 non-OPiv substrates for testing, as shown in Figure 8(c).

- **The real-world electronic laboratory notebook (ELN) dataset** [40] was created for Buchwald-Hartwig reactions from electronic laboratory notebooks, including 750 reactions. The structural diversity of the real-world ELN dataset is much higher than that of the HTE datasets. The random split also simulates an out-of-sample scenario.

- **The asymmetric N,S-acetal formation using CPA catalysts (N,S-acetal) dataset** [42] included combinatorial variations of CPA catalysts, N-acyl imines, and thiols, resulting in a total of 1,075 reactions. We choose a set of catalysts (test-cat) as the pivot to construct the out-of-sample split condition, including 24 catalysts for training and 19 for testing, as shown in Figure 8(d).

- **The nanomole-scale reactivity evaluation of C-heteroatom-coupling reactions (C-heteroatom) dataset** [43] was performed by an automated high-throughput screening on a nanomole scale, yielding 1,536 reactions. We choose a set of catalysts (test-cat) as the pivot to construct the out-of-sample split condition, where the split uses 12 catalysts for training and 4 for testing, as shown in Figure 8(e).

- **USPTO_TPL dataset** [8] labels were generated by extracting the 1,000 most common templates from the USPTO dataset, containing 445,115 reactions.

Figure 7 presents distributions of the number of symmetric difference and union shingles per reaction for each dataset.

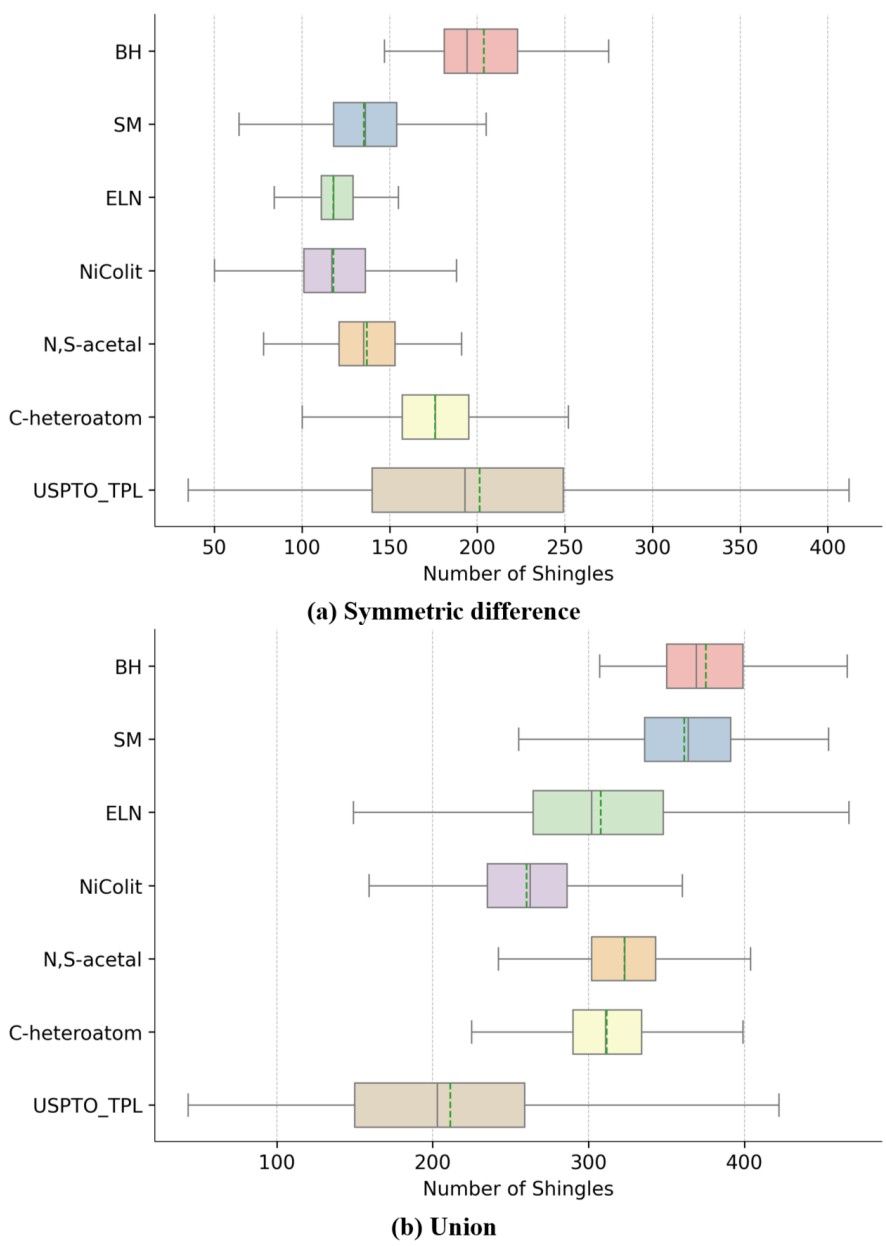

Figure 7: **Distributions of the number of shingles per reaction**. (a) The symmetric difference of shingles removes many duplicate shingles in the intersection. (b) The union of shingles from reactants and products.

## C  Shingle generation

In practice, we also treat ring structures within molecules as shingles, as they capture rich and informative substructures. We set the max radius to 3. To balance expressiveness and computational efficiency, we consider up to 280 shingles per reaction and 100 shingles per molecule, allowing each unique shingle to appear up to 10 times. The per-reaction bound should be adapted to the downstream dataset at hand. The per-molecule bound prevents rare cases where one molecule generates an excessive number of shingles and dominates the representation. The generation of the symmetric difference shingle set is depicted in Algorithm 1.

Note that for the USPTO_TPL dataset, we generate shingles solely from reactants. This choice is motivated by the observation that the symmetric difference of substructures between reactants and

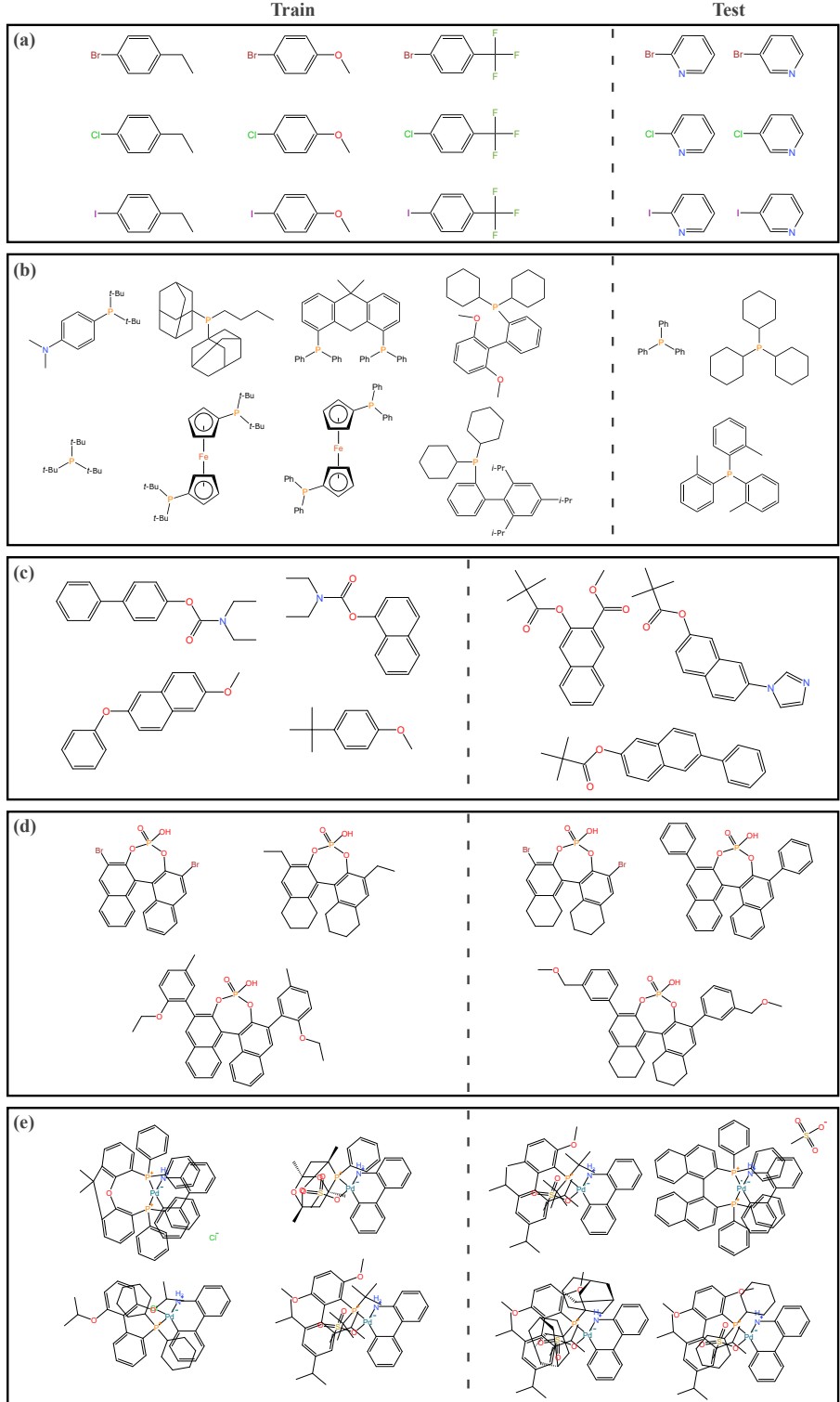

Figure 8: **Example molecules used in out-of-sample splits**. (a) Reactant-based for the BH dataset. (b) Ligand-based for the SM dataset. (c) Substrate-based for the NiCOlit dataset. (d) Catalyst-based for the N,S-acetal dataset. (e) Catalyst-based for the C-heteroatom dataset.

products is often minimal. In practice, we can apply the symmetric difference, reactants-only, and union strategies, and select the one that yields the best performance.

**Algorithm 1** Pipeline to extract symmetric difference shingles

---

**Input:** the reaction $\mathcal{R} = \{\mathcal{M}^r; \mathcal{M}^p\}$ divided into reactants and products, and the maximum radius $r_{max}$ for shingles
**Output:** symmetric difference shingle set $S_\mathcal{R}$ between reactants and products

```
 1: S_react ← {}                                           ▷ Initialize reactants shingle set
 2: S_prod ← {}                                            ▷ Initialize products shingle set
 3: for m in M^r do                                        ▷ Enumerate molecules and atoms
 4:     for v in m do
 5:         for r ← 1 to r_max do
 6:             calculate and add shingles S^(r)(v) to S_react    ▷ Generate shingles for reactants
 7:         end for
 8:     end for
 9: end for
10: for m in M^p do                                        ▷ Enumerate molecules and atoms
11:     for v in m do
12:         for r ← 1 to r_max do
13:             calculate and add shingles S^(r)(v) to S_prod     ▷ Generate shingles for products
14:         end for
15:     end for
16: end for
17: S_R = (S_react \ S_prod) ∪ (S_prod \ S_react)          ▷ Compute symmetric difference
18: return S_R
```

---

## D   Pre-training settings

We introduce three pseudo-reaction-type classification tasks as pre-training objectives. The underlying intuition is that similar chemical structures tend to exhibit consistent semantic behavior across various reactions. Specifically, we employ the DRFP [31] with default parameters, which is a 1024-length one-hot descriptor as the reaction fingerprint. These reaction fingerprint sequences can be used for clustering, where shorter distances between fingerprints indicate a higher likelihood of belonging to the same cluster. We apply $K$ means clustering by scikit-learn [61] with different values of $K$ (100, 1,000, 4,000, see Figure 9 concerning selection of $K$) to cluster reactions to obtain clusters with different granularities from coarse-grained to fine-grained. Subsequently, we use three classification heads with output sizes of 100, 1,000, and 4,000, respectively, to predict the pseudo labels associated with each cluster. This approach leverages the structural consistency of chemical reactions, enabling the model to learn more robust and transferable representations.

## E   Comparison with standard cross-attention architecture

We implement a cross-attention baseline that explicitly models interactions between reactants and products. In this architecture, product embeddings serve as queries, while reactant embeddings are used as keys and values within a standard multi-head cross-attention module [33]. The model consists of 4 cross-attention layers, 64 attention heads, and a hidden dimension of 512, and it shares the same molecular encoders as ReaDISH.

We evaluate this architecture for six datasets to compare it with ReaDISH. As shown in Table 4, the cross-attention baseline consistently underperforms our proposed model. This result suggests that simple cross-attention treats inter-molecular interactions uniformly, lacking chemical or geometric guidance. In contrast, ReaDISH explicitly incorporates both inter- and intra-molecular relationships through structural distances, edge types, and symmetric-difference shingles, resulting in more chemically grounded representations.

Table 4: Comparison between ReaDISH and the cross-attention model for six datasets.

| Model | BH | SM | NiCOlit | N,S-acetal | C-heteroatom | ELN (OOS) |
|---|---|---|---|---|---|---|
| Cross-attention | $0.946 \pm 0.005$ | $0.858 \pm 0.009$ | $0.431 \pm 0.007$ | $0.844 \pm 0.005$ | $0.705 \pm 0.076$ | $0.156 \pm 0.006$ |
| ReaDISH | $0.976 \pm 0.001$ | $0.886 \pm 0.008$ | $0.539 \pm 0.024$ | $0.916 \pm 0.007$ | $0.763 \pm 0.026$ | $0.230 \pm 0.047$ |

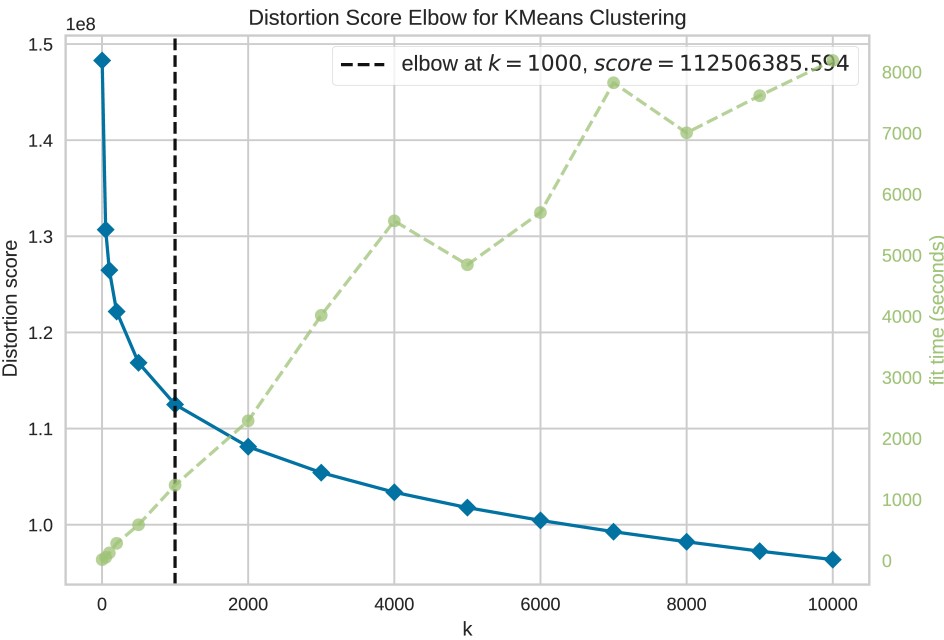

Figure 9: **The elbow for the $K$ means algorithm**. We use the distortion score to find the elbow and choose $K = 100, 1,000, 4,000$ for reaction clustering.

## F  Implementation details

We extract the results of the baseline models from the existing literature as much as possible, and otherwise reproduce them based on publicly available code. Note that UAM applies contrastive pre-training on the whole dataset in downstream tasks, which may introduce data leakage. Hence, we randomly initialize the model weights to fine-tune the model during reproduction. All methods are tested on (1) the same ten random splits and (2) the same out-of-sample split across five random runs to ensure fair comparisons, with the average results reported.

Table 5 and 6 present hyper-parameters used in ReaDISH during pre-training and fine-tuning, respectively. We use Pytorch [62] with the Adam [63] optimizer and the cosine learning rate decay strategy for training. All experiments are executed on 4 NVIDIA RTX3090 GPUs. Pre-training the model takes about 8 hours, whereas fine-tuning on downstream datasets requires up to 1 hour. Generating shingles for the pre-training dataset takes around 7 hours using 50 CPU threads, while processing a downstream dataset completes in under 4 minutes. ReaDISH contains about 16.3M trainable parameters.

Table 5: Parameters during pre-training.

| Parameters | Value | Parameters | Value |
|---|---|---|---|
| Number of encoder layers | 4 | Number of encoder attention heads | 64 |
| Encoder FFN embedding dimension | 2048 | Encoder embedding dimension | 512 |
| Batch size | 64 | Initial learning rate | 5e-5 |
| Max epochs | 3 | Minimum learning rate | 5e-6 |
| Warmup steps | 2000 | Warmup learning rate | 1e-6 |
| Dropout rate | 0.1 | Number of Gaussian kernels | 128 |

## G  Dependence of the prediction performance on the size of the training data

We scale training size on BH and SM benchmarks to measure performance changes, using subsets of 5%, 10%, 20%, 30%, 50%, and 70% while testing on the full 30% test split. Results in Table 7 and

Table 6: Search space of parameters during fine-tuning.

| Parameters | Search space |
|---|---|
| Max epochs | 150, 200 |
| Batch size | 64, 128 |
| Initial learning rate | 5e-3, 1e-3, 5e-4 |
| Minimum learning rate | 5e-4, 1e-4, 5e-5 |

Figure 10 show that predictive accuracy ($R^2$) steadily improves without saturation, suggesting further gains with larger datasets. But the rate of increase slows down.

Table 7: Performance regarding training size on the BH and SM datasets.

| Dataset | 5% | 10% | 20% | 30% | 50% | 70% |
|---|---|---|---|---|---|---|
| BH | $0.713 \pm 0.017$ | $0.808 \pm 0.017$ | $0.874 \pm 0.016$ | $0.919 \pm 0.006$ | $0.955 \pm 0.002$ | $0.976 \pm 0.001$ |
| SM | $0.508 \pm 0.018$ | $0.649 \pm 0.008$ | $0.751 \pm 0.010$ | $0.810 \pm 0.001$ | $0.851 \pm 0.010$ | $0.886 \pm 0.008$ |

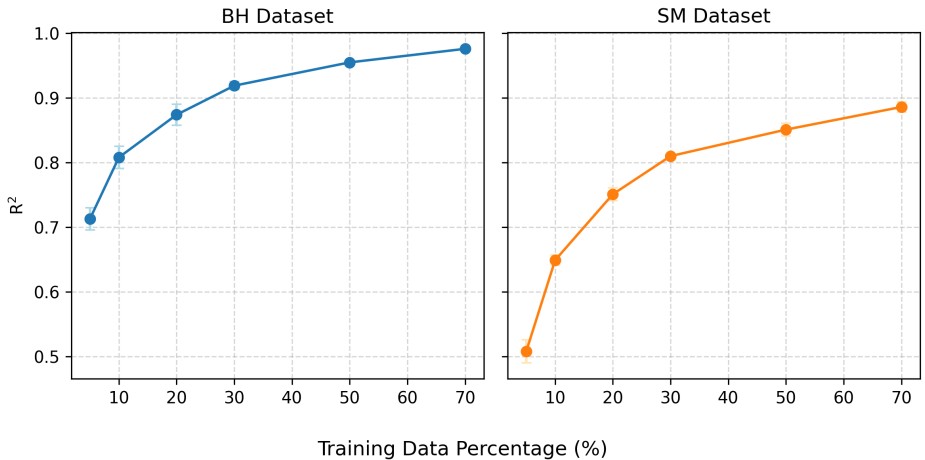

Figure 10: **The learning curve regarding training size** for the BH (left) and SM (right) datasets.

## H  Additional experimental results

The full experimental results are detailed in Table 8 and Table 9.

## I  Limitations

While ReaDISH effectively models chemical reactions through the symmetric difference of molecular shingles, one limitation is its computational overhead. In cases where all substructures are considered, the number of shingles can exceed the number of atoms, leading to increased memory consumption and longer training times. We mitigate this challenge by setting an upper bound on the number of shingles per reaction, balancing expressiveness with efficiency. Future work could explore adaptive pruning strategies to further reduce resource demands without compromising predictive performance.

## J  Impact statements

Advances in chemical reaction prediction hold the potential to accelerate scientific discovery in areas such as drug development, materials science, and sustainable chemistry. However, enhanced

Table 8: Results under random splits. Bold entries highlight the best performance.

| Method | BH | | | SM | | | NiCOlit | | |
|---|---|---|---|---|---|---|---|---|---|
| | MAE | RMSE | $R^2$ | MAE | RMSE | $R^2$ | MAE | RMSE | $R^2$ |
| YieldBERT | $3.99 \pm 0.15$ | $6.01 \pm 0.27$ | $0.951 \pm 0.005$ | $8.13 \pm 0.34$ | $12.07 \pm 0.46$ | $0.815 \pm 0.013$ | $19.96 \pm 1.95$ | $27.23 \pm 1.38$ | $0.353 \pm 0.070$ |
| YieldBERT-DA | $3.09 \pm 0.12$ | $4.80 \pm 0.26$ | $0.969 \pm 0.004$ | $6.60 \pm 0.27$ | $10.52 \pm 0.48$ | $0.859 \pm 0.012$ | $19.53 \pm 1.82$ | $26.53 \pm 1.28$ | $0.373 \pm 0.060$ |
| UA-GNN | $2.92 \pm 0.06$ | $4.43 \pm 0.09$ | $0.974 \pm 0.001$ | $6.12 \pm 0.22$ | $9.47 \pm 0.46$ | $0.886 \pm 0.010$ | $16.19 \pm 0.26$ | $22.23 \pm 0.43$ | $\mathbf{0.550 \pm 0.017}$ |
| UAM | $\mathbf{2.89 \pm 0.06}$ | $\mathbf{4.36 \pm 0.10}$ | $\mathbf{0.976 \pm 0.001}$ | $\mathbf{6.04 \pm 0.18}$ | $\mathbf{9.23 \pm 0.40}$ | $\mathbf{0.888 \pm 0.009}$ | $16.51 \pm 0.87$ | $22.91 \pm 1.12$ | $0.515 \pm 0.053$ |
| ReaMVP | $3.11 \pm 0.07$ | $4.63 \pm 0.14$ | $0.971 \pm 0.002$ | $6.59 \pm 0.20$ | $10.37 \pm 0.42$ | $0.864 \pm 0.010$ | $16.21 \pm 0.87$ | $22.61 \pm 1.12$ | $0.541 \pm 0.053$ |
| ReaDISH | $2.98 \pm 0.05$ | $\mathbf{4.36 \pm 0.09}$ | $\mathbf{0.976 \pm 0.001}$ | $6.09 \pm 0.18$ | $9.55 \pm 0.39$ | $0.886 \pm 0.008$ | $\mathbf{15.36 \pm 0.28}$ | $\mathbf{21.89 \pm 0.57}$ | $0.539 \pm 0.024$ |

| Method | N,S-acetal | | | C-heteroatom | | | USPTO_TPL | | |
|---|---|---|---|---|---|---|---|---|---|
| | MAE | RMSE | $R^2$ | MAE | RMSE | $R^2$ | ACC | MCC | CEN |
| YieldBERT | $0.16 \pm 0.01$ | $0.23 \pm 0.02$ | $0.861 \pm 0.015$ | $1.23 \pm 0.14$ | $2.61 \pm 0.11$ | $0.664 \pm 0.057$ | $0.989$ | $0.989$ | $0.006$ |
| YieldBERT-DA | $0.16 \pm 0.01$ | $0.22 \pm 0.01$ | $0.889 \pm 0.017$ | $1.03 \pm 0.12$ | $2.58 \pm 0.06$ | $0.709 \pm 0.043$ | $0.990$ | $0.990$ | $0.006$ |
| UA-GNN | $0.15 \pm 0.00$ | $0.21 \pm 0.01$ | $0.905 \pm 0.007$ | $0.90 \pm 0.08$ | $2.30 \pm 0.17$ | $0.751 \pm 0.038$ | $0.991$ | $0.991$ | $\mathbf{0.005}$ |
| UAM | $0.19 \pm 0.01$ | $0.26 \pm 0.02$ | $0.859 \pm 0.012$ | $1.04 \pm 0.11$ | $2.65 \pm 0.10$ | $0.718 \pm 0.047$ | $0.954$ | $0.954$ | $0.250$ |
| ReaMVP | $0.14 \pm 0.01$ | $0.21 \pm 0.01$ | $0.904 \pm 0.012$ | $0.90 \pm 0.09$ | $2.57 \pm 0.11$ | $0.726 \pm 0.043$ | $0.990$ | $0.990$ | $0.006$ |
| ReaDISH | $\mathbf{0.14 \pm 0.00}$ | $\mathbf{0.20 \pm 0.01}$ | $\mathbf{0.916 \pm 0.007}$ | $\mathbf{0.87 \pm 0.04}$ | $\mathbf{1.94 \pm 0.10}$ | $\mathbf{0.763 \pm 0.026}$ | $\mathbf{0.992}$ | $\mathbf{0.992}$ | $\mathbf{0.005}$ |

Table 9: Results under out-of-sample splits. Bold entries highlight the best performance.

| Method | BH (reactant-based) | | | SM (ligand-based) | | | NiCOlit (substrate-based) | | |
| --- | --- | --- | --- | --- | --- | --- | --- | --- | --- |
| | MAE | RMSE | $R^2$ | MAE | RMSE | $R^2$ | MAE | RMSE | $R^2$ |
| YieldBERT | 17.44 ± 1.01 | 23.90 ± 1.13 | 0.350 ± 0.060 | 14.85 ± 0.36 | 19.59 ± 0.39 | 0.469 ± 0.021 | 15.53 | 21.91 | 0.463 |
| YieldBERT-DA | 18.41 ± 0.48 | 26.30 ± 0.33 | 0.215 ± 0.020 | 15.78 ± 0.24 | 19.64 ± 0.26 | 0.467 ± 0.014 | 15.05 | **21.21** | 0.483 |
| UA-GNN | 16.95 ± 0.21 | 24.82 ± 0.63 | 0.301 ± 0.035 | 15.59 ± 0.36 | 20.48 ± 0.39 | 0.420 ± 0.022 | **14.82** | 22.19 | 0.563 |
| UAM | 16.99 ± 0.55 | 21.78 ± 1.13 | 0.458 ± 0.051 | 14.46 ± 0.37 | 18.63 ± 0.41 | 0.520 ± 0.029 | 17.64 | 22.99 | 0.446 |
| ReaMVP | 17.17 ± 0.83 | **21.40 ± 1.15** | 0.479 ± 0.056 | **13.90 ± 0.29** | 18.36 ± 0.35 | 0.534 ± 0.018 | 15.01 | 21.43 | 0.489 |
| ReaDISH | **16.29 ± 0.30** | 21.76 ± 0.80 | **0.480 ± 0.032** | 14.22 ± 0.33 | **18.05 ± 0.37** | **0.549 ± 0.019** | 15.09 | 21.91 | **0.574** |

| Method | ELN | | | N,S-acetal (catalyst-based) | | | C-heteroatom (catalyst-based) | | |
| --- | --- | --- | --- | --- | --- | --- | --- | --- | --- |
| | MAE | RMSE | $R^2$ | MAE | RMSE | $R^2$ | MAE | RMSE | $R^2$ |
| YieldBERT | 21.98 ± 2.27 | 27.67 ± 2.14 | 0.151 ± 0.102 | 0.258 | 0.361 | 0.735 | 2.11 | 3.70 | 0.309 |
| YieldBERT-DA | 21.58 ± 2.19 | 26.97 ± 1.98 | 0.171 ± 0.112 | 0.239 | 0.340 | 0.764 | 1.96 | 3.58 | 0.351 |
| UA-GNN | 20.64 ± 1.13 | 26.50 ± 1.03 | 0.203 ± 0.054 | 0.239 | 0.329 | 0.780 | 1.77 | 3.46 | 0.394 |
| UAM | 22.10 ± 1.35 | 26.65 ± 1.78 | 0.179 ± 0.050 | 0.305 | 0.429 | 0.630 | 2.01 | 3.62 | 0.346 |
| ReaMVP | 20.69 ± 1.33 | 26.36 ± 1.29 | 0.212 ± 0.057 | 0.234 | 0.342 | 0.762 | 2.04 | 3.54 | 0.361 |
| ReaDISH | 20.82 ± 1.01 | **25.99 ± 1.12** | **0.230 ± 0.047** | **0.226** | **0.327** | **0.790** | 2.02 | **3.42** | **0.409** |

predictive models could be misused to facilitate the design of harmful substances, including toxic chemicals or controlled compounds. This raises ethical and security concerns regarding dual-use applications. Additionally, widespread adoption of automated chemical design tools may shift traditional roles in chemical research, with potential implications for education and workforce development.

