# OpenReview forum: "Reaction Prediction via Interaction Modeling of Symmetric Difference Shingle Sets"
_NeurIPS.cc/2025/Conference — NeurIPS 2025 poster_

### Official Review · Reviewer_FGGr · 2025-06-21

**Clarity:** 3
**Significance:** 3
**Originality:** 3
**Rating:** 5
**Confidence:** 4

**Summary:**

The authors present ReaDISH, a reaction prediction model that addresses permutation sensitivity and substructure interaction modeling by introducing two key innovations: a symmetric difference of molecular shingles and an interaction-aware attention mechanism integrating geometric and chemical distances. The method achieves improved accuracy and generalization across multiple reaction prediction tasks, especially under permutation and out-of-sample settings.

**Questions:**

- Can the authors provide more details on the computational efficiency of the shingle generation process?
- Is the symmetric difference between reactant and product shingles always well-defined and non-empty? If not, how are such cases handled？
- How does ReaDISH perform with large molecules or complex reaction datasets, especially in terms of computation time and memory? How much additional GPU memory and runtime does the union-based variant consume compared to symmetric difference, and does it lead to better predictive performance?
- While several tasks are evaluated, could the authors comment on task-specific strengths or weaknesses? For example, is ReaDISH particularly strong in stereoselectivity due to interaction modeling?

**Ethical Concerns:**

["NO or VERY MINOR ethics concerns only"]

**Final Justification:**

The author's response addressed my prior concerns. I maintain the original positive rating.

**Limitations:**

yes

**Quality:**

3

**Strengths And Weaknesses:**

## Strengths

- Well-written and easy to follow.
- Novel chemical reaction modeling method.
- Extensive experiments are conducted to demonstrate the efficacy of ReaDISH.

## Weaknesses

- The paper lacks a detailed analysis of ReaDISH's computational complexity and resource requirements.
- As acknowledged, the model's scalability is a key limitation. Handling a large number of shingles requires higher computational costs for complex reactions.

---

> ### Author Rebuttal · Authors · 2025-07-26
>
> Thank you for your time and constructive feedback. In the following, we have addressed your concerns and questions point-by-point.
>
> > **Q1 & W2: Shingle generation efficiency**
>
> We understand the reviewer's concerns regarding the efficiency of shingle generation. Shingle generation and pairwise interaction computation are highly efficient: even for large molecules, processing a downstream dataset takes under 4 minutes. For our large-scale pretraining corpus of 3.9M reactions, generation requires ~7 hours on 50 CPU threads. We will explicitly mention the computational cost in the appendix.
>
> > **Q2: Validity of symmetric difference**
>
> We understand the reviewer's concerns regarding the validity of symmetric difference. The symmetric difference between reactant and product shingles is **almost always non‑empty**, as reactions involve meaningful structural changes. Rare empty cases arise from: (1) **Simple reactions**: The structural change is minimal, and the union of reactant and product shingles can still provide a meaningful representation. (2) **Data errors**: Some reactions in the pretraining datasets are incorrectly recorded (e.g., A -> A + B). These are identified and removed during preprocessing. We emphasize that such cases do not appear during fine-tuning.
>
> > **Q3 & W1: Computational efficiency analysis**
>
> We understand the reviewer's concerns regarding computational efficiency. ReaDISH is computationally efficient across datasets. Shingle generation remains under 4 minutes for thousands of reactions, and the larger benchmark (e.g., BH) requires ~1 hour of training time compared to ~0.5 hour for smaller datasets (e.g., ELN). The union‑based variant **increases memory by ~3 times** on large datasets (e.g., SM) and **does not improve accuracy** due to redundant shingles. We will explicitly mention the computational cost in the appendix for clarity.
>
> > **Q4: Task-specific strengths or weaknesses**
>
> Thank you very much for your interest in our method. By modeling the symmetric difference of shingles, ReaDISH captures **structural transformations** critical for reaction prediction, particularly for heteroatom‑centered and stereochemically sensitive tasks. We observe 1.5% (C-heteroatom) and 1.8% (ELN) gains in out‑of‑sample evaluations, demonstrating its suitability for such chemical changes.

---

> ### Author Response · Authors · 2025-08-07
> **Official Comment by Authors**
>
> Dear Reviewer,
>
> I hope this message finds you well. As the discussion period is approaching its end with fewer than three days remaining, we want to ensure that we have addressed all of your concerns to your satisfaction. If you have any additional comments or suggestions you would like us to consider, please don't hesitate to let us know. Your feedback is highly valued, and we are eager to address any remaining issues to improve our work further.
>
> Thank you again for your time and thoughtful engagement throughout the review process. Wishing you a wonderful day!

---

> ### Comment · Area_Chair_oWBd · 2025-08-09
>
> Reviewer FGGr
> Submitting a “Mandatory Acknowledgement” without leaving any comments for the authors in the discussion is not allowed. Could you provide feedback on the authors’ responses to your review comments? AC

---

> > ### Comment · Reviewer_FGGr · 2025-08-09
> > **Follow-up**
> >
> > Dear AC,
> >
> > The author's response addressed my prior concerns. I maintain the original positive rating.
> >
> > I originally include this comments in the final comment, and I was not aware that that cannot be seen by the authors.

---

> > > ### Author Response · Authors · 2025-08-09
> > >
> > > We are pleased that your prior concerns have been addressed and sincerely appreciate your positive decision. Your thoughtful feedback and engagement throughout the review process have been invaluable in improving the quality and clarity of our work. We will incorporate these changes and clarifications into the final version. Should you have any further questions or suggestions, we would be happy to continue the discussion.

---

### Official Review · Reviewer_RXFe · 2025-06-29

**Clarity:** 3
**Significance:** 3
**Originality:** 3
**Rating:** 4
**Confidence:** 4

**Summary:**

The article "Reaction Prediction via Interaction Modeling of Symmetric Difference Shingle Sets" describes the development of an architecture for reaction outcome prediction. The authors show interesting performance on relevant benchmark datasets. However, several details regarding the model architecture and its performance characterization are unclear. Please find my detailed comments below that should be addressed in order for this submission to be accepted.

**Questions:**

Major aspects:
Page 14, Section B: The authors state that they sample 100 conformers for each molecule and sample one of them. It is not clear how this one conformer is chosen. How critical is the necessity to sample 100 conformers? Additionally, I think the authors should demonstrate the sensitivity of their model to the particular conformer selected. This prompts the question whether the case illustrated in Figure 6 with permutations and token orderings was done under the assumption of exactly identical Cartesian coordinates or whether this also included the re-generation of Cartesian coordinates, which can lead to different Cartesian coordinates and thus different embeddings and predictions.

It is not clear from the text whether the performance comparisons shown in Table 1 and Table 2 were done on the training set or the test set. In case they were done on the test set, identifying the best model corresponds to a hyperparameter optimization, and the accuracy of the model cannot be really estimated. This would require a separate split.

Page 16, Section C: The authors state that the max radius of the shingles was set to 3. Why was this done? I think it is critical to show a dependency of the model performance on shingle size. Additionally, the authors use an upper bound on the number of shingles per reaction. How critical is this parameter? Are there any strategies for when there are more than 100 potentially relevant shingles to select shingles that are meaningful to the reaction?

The ablation study by the authors summarized in Table 2 appears to show that, apart from the pairwise interaction treatment, the other components of the model only seem to marginally improve prediction performance. In addition, pre-training seems to have the lowest impact of all. I would like to see additional ablation studies, especially one that entirely removes the use of shingles from the model.

It is not clear whether the authors performed any hyperparameter optimization. I think this needs to be mentioned explicitly. If hyperparameter optimization was performed, it should be mentioned how it was performed.

I think it is important to also show a learning curve for the new architecture. With that, I mean a dependence of the prediction performance on the size of the training data. This illustrates in what dataset size regime the model learns and it also indicates whether the model might continue learning if provided with more data.

I think that the prediction performance should be compared to more alternative approaches. In particular, at least for some of the subsets used for training (e.g., BH, SM, etc.), other models have been developed. The corresponding performance should provided and compared to the current model. This is important even when data splits are not the exact same between different approaches.

Some details regarding shingle generation are not clear based on the provided text. This is an issue as no code has been provided for the submission. For instance, it is stated that ring structures within molecules are shingles but how exactly they are extracted and what the additional environment apart from the ring structures is not clear. I think more details need to be provided for that.

The authors apply the model to various datasets that constitute, among others, yield prediction and enantioselectivity prediction. Does this mean that the model architecture is trained on each dataset separately to reach the described performance or is there one model trained on all the datasets?

I think Tables 1 and 2 should not only provide results for BH and SM but also for the other datasets. This allows for a more balanced evaluation of model performance. Some of these results are provided in the Appendix but I think they should be provided in the main text.

Minor aspects:
Table 1: The best value is not always bolded.

**Ethical Concerns:**

["NO or VERY MINOR ethics concerns only"]

**Final Justification:**

Based on the extensive discussion with the authors, I have decided to raise my review score to 4. I think the authors addressed all of my concerns and were able to resolve most of them. In my opinion, the main outstanding points that prevented an even higher score were the absence of an appropriate estimate of final model performance, the fact that ReaDISH, at least for some very well studied literature baselines that allow for comparison to a wide range of methods, likely reaches state of the art but does not outperform the current state of the art, and the fact that ReaDISH, while a powerful framework, is highly sensitive with respect to key hyperparameters, making it somewhat less straightforward to apply effectively in practice than some of the current baseline approaches in the field.

**Limitations:**

Yes.

**Paper Formatting Concerns:**

None.

**Quality:**

3

**Strengths And Weaknesses:**

I think the quality is good as the datasets used for model evaluation are diverse and good statistical metrics were used.
I think the clarity is fair but requires further improvement as there are several points that require further clarification.
I think the significance is fair as the model architecture improves performance relative to existing baselines. However, some baseline models are missing which hampers significance.
I think originality is good as the model architecture provides several innovative improvements compared to baseline models.

---

> ### Author Rebuttal · Authors · 2025-07-26
>
> Thank you for taking your valuable time and providing us with valuable feedback. In the following, we have addressed your concerns and questions point-by-point.
>
> > **Q1: Conformer generation and sensitivity**
>
> We understand the reviewer's concerns regarding conformers. We generate 100 conformers to approximate plausible 3D geometries, cluster them using the RDKit Butina algorithm (rdkit.ML.Cluster.Butina, threshold 0.25), and sample a conformer from the representative clusters with probabilities proportional to their sizes. This procedure ensures that the chosen conformer is both **structurally plausible (reflecting common conformations)** and **diverse enough** to avoid biasing the model toward a single geometry. To assess sensitivity, we have repeated training and evaluation five times **using independently sampled conformers** and performance variations (standard deviations) are minimal (see Tables 6 and 7 in the appendix), indicating robustness to conformer choice. Regarding Figure 6, we clarify that the permutation and token-ordering analysis is performed **using identical conformers**, and we will explicitly state this assumption in the revised version.
>
> > **Q2: Hyperparameter optimization**
>
> We understand the reviewer's concerns regarding optimizing hyperparameters. All results reported in tables and figures are based on the test set, with hyperparameters selected using the training set following prior work to ensure fair comparison. While we acknowledge that an additional validation set could further guard against overfitting, our setup ensures **fair comparison** with existing methods.
>
> > **Q3: Radius and number of shingles**
>
> We understand the reviewer's concerns regarding the detailed settings of shingles. (1) We set the radius to 3 as the best trade-off between performance and efficiency, as prior works in cheminformatics do [1,2]. We have **added ablation studies with radius** and will update in the revised version, as shown in the table below. Radius 4 performs the worst, likely due to generating too many shingles, leading to redundancy and increased complexity, as well as higher runtime and memory costs. Radius 2 is slightly worse than radius 3. (2) We apologize for the typo of "100 shingles per reaction". Actually, our implementation considers up to **280 shingles per reaction** and 100 shingles per molecule. The per-molecule bound prevents rare cases where one molecule generates an excessive number of shingles and dominates the representation. The per-reaction bound functions similarly to a maximum sequence length in Transformers, preventing rare, large reactions from causing OOM issues. Figure 7 in the appendix shows the shingle distribution, confirming that the bound comfortably covers nearly all reactions. (3) We currently do not apply additional selection strategies for reactions exceeding this bound and have noted this as a future direction in the limitations section.
>
> | Methods    | BH (OOS)         |                  |                   | SM (OOS)         |                  |                   |
> | ---------- | ---------------- | ---------------- | ----------------- | ---------------- | ---------------- | ----------------- |
> |            | MAE              | RMSE             | R$^2$                | MAE              | RMSE             | R$^2$                |
> | r=2        | $16.50 \pm 0.15$ | $22.11 \pm 0.25$ | $0.465 \pm 0.014$ | $14.41\pm0.33$   | $18.57\pm0.44$   | $0.523\pm0.228$   |
> | r=4        | $17.91 \pm 0.40$ | $23.90 \pm 0.01$ | $0.421 \pm 0.006$ | $14.88\pm0.56$   | $18.89\pm0.82$   | $0.505\pm0.043$   |
> | r=3 (ours) | $16.29 \pm 0.30$ | $21.76 \pm 0.80$ | $0.480 \pm 0.032$ | $14.22 \pm 0.33$ | $18.05 \pm 0.37$ | $0.549 \pm 0.019$ |
>
> > **Q4: More ablation studies**
>
> We understand the reviewer's concerns regarding ablation studies. Beyond the radius ablation (see Q3), we have **conducted an additional ablation removing shingles and using only atoms**, and will update in the revised version. In this case, symmetric difference cannot be applied (atoms remain unchanged between reactants and products), so we adopt a union-based representation for comparison. Results in the table below show that this atom-only variant performs worse than the shingle-based model, confirming that **local substructures are critical** for modeling reaction transformations.
>
> | Methods          | BH (OOS)         |                  |                   | SM (OOS)         |                  |                   |
> | ---------------- | ---------------- | ---------------- | ----------------- | ---------------- | ---------------- | ----------------- |
> |                  | MAE              | RMSE             | R$^2$                | MAE              | RMSE             | R$^2$                |
> | w/o shingle      | $17.97\pm0.26$   | $23.72\pm0.18$   | $0.409\pm0.015$   | $14.51\pm0.32$   | $18.85\pm0.47$   | $0.508\pm0.024$   |
> | w shingle (ours) | $16.29 \pm 0.30$ | $21.76 \pm 0.80$ | $0.480 \pm 0.032$ | $14.22 \pm 0.33$ | $18.05 \pm 0.37$ | $0.549 \pm 0.019$ |
>
> > **Q5: Performance regarding training size**
>
> We understand the reviewer's concerns regarding training size. We have scaled training size on BH and SM benchmarks and will update in the revised version, **using subsets of 5%, 10%, 20%, 30%, 50%, and 70% while testing on the full 30% test split**. Results in the table below show that predictive accuracy (R$^2$) steadily improves without saturation, suggesting further gains with larger datasets. But the rate of increase has slowed down. We will add this analysis to the appendix.
>
> | Dataset | 5%                | 10%             | 20%             | 30%             | 50%             | 70%             |
> | ------- | ----------------- | --------------- | --------------- | --------------- | --------------- | --------------- |
> | BH      | $0.713 \pm 0.173$ | $0.808\pm0.172$ | $0.874\pm0.168$ | $0.919\pm0.006$ | $0.955\pm0.002$ | $0.976\pm0.001$ |
> | SM      | $0.508\pm0.018$   | $0.649\pm0.008$ | $0.751\pm0.010$ | $0.810\pm0.001$ | $0.851\pm0.010$ | $0.886\pm0.008$ |
>
> > **Q6: More baseline comparison**
>
> We understand the reviewer's concerns regarding baselines. We have **added Equiformer** [3], an equivariant graph attention model, as a baseline and will update in the revised version. Although originally designed for single-molecule representation, we adapt it for reactions by merging reactants and products into a unified graph, computing edges within each molecule. Results (R$^2$) in the table below show that **our model outperforms Equiformer** on various benchmarks, underscoring the advantages of our symmetric-difference shingle design.
>
> | Model (random)            | BH                | SM                | NiCOlit           | N,S-acetal        | C-heteroatom      |                   |
> | ------------------------- | ----------------- | ----------------- | ----------------- | ----------------- | ----------------- | ----------------- |
> | Equiformer                | $0.958 \pm 0.004$ | $0.847 \pm 0.003$ | $0.497 \pm 0.006$ | $0.875 \pm 0.002$ | $0.713 \pm 0.004$ |                   |
> | ReaDISH                   | $0.976\pm0.001$   | $0.886\pm0.008$   | $0.539 \pm 0.024$ | $0.916 \pm 0.007$ | $0.763 \pm 0.026$ |                   |
> | **Model (out-of-sample)** | **BH**            | **SM**            | **NiCOlit**       | **N,S-acetal**    | **C-heteroatom**  | **ELN**           |
> | Equiformer                | $0.429 \pm 0.005$ | $0.449 \pm 0.014$ | $0.442$           | $0.731$           | $0.321$           | $0.178 \pm 0.005$ |
> | ReaDISH                   | $0.480 \pm 0.032$ | $0.549 \pm 0.019$ | $0.574$           | $0.790$           | $0.409$           | $0.230 \pm 0.047$ |
>
> > **Q7: Shingle generation details**
>
> We understand the reviewer's concerns regarding generating shingles. We clarify that **code has already been submitted in the initial submission** to ensure reproducibility. Each shingle is defined by a fixed-radius local environment plus whole rings detected via RDKit (AllChem.GetSymmSSSR).
>
> > **Q8: Training separately or not**
>
> We train **separately per dataset** to optimize for task-specific objectives (e.g., yield regression vs. reaction classification). While multi-task pretraining is possible, our goal is not to build a foundation model but to show the benefits of shingles compared to task-specific baselines.
>
> > **Q9: More results in the main text**
>
> We agree that presenting more comprehensive results in the main text would improve clarity. We will move additional results from the appendix to the main text in the revised version. Performance trends remain consistent across datasets, but paper **space constraints** previously limited the inclusion.
>
> > **Q10: A typo error**
>
> We thank the reviewer for identifying it. The missing bold formatting is a typographical error and will be corrected in the revised version.
>
> ---
>
> **References**
>
> [1] A. Capecchi, D. Probst, and J. L. Reymond. One molecular fingerprint to rule them all: drugs, biomolecules, and the metabolome. *Journal of Cheminformatics*, 12:1–15, 2020.
>
> [2] M. Orsi and J. L. Reymond. One chiral fingerprint to find them all. *Journal of Cheminformatics*, 16(1):53, 2024.
>
> [3] Liao, Y. L. & Smidt, T. Equiformer: Equivariant Graph Attention Transformer for 3D Atomistic Graphs. *ICLR 2023*.

---

> > ### Comment · Reviewer_RXFe · 2025-08-01
> >
> > I thank the authors for responding to the concerns raised in my review. Overall, I think several of the concerns are adressed but some of my concerns are not addressed sufficiently. Here is my response to the rebuttal:
> >
> > Q1: Thank you for providing this additional information, this answers my concern raised in the original review sufficiently.
> >
> > Q2: Thank you for this clarification. I acknowledge this allows for a fair comparison but maintain my remark that overall model performance cannot be assessed reliably that way.
> >
> > Q3: This additional information is critical and gives some insight regarding the radius hyperparameter. Regarding the upper bound on the number of shingles. While the rebuttal provides some insight into why a certain number might be chosen, I do not think that the authors answer my question about how critical this number is for maintaining prediction performance.
> >
> > Q4: Thank you for providing this additional critical baseline confirming the positive effect of shingles.
> >
> > I would like to note that it seems the authors have overlooked my concern regarding absence of any information about whether and how hyperparameter optimization was performed.
> >
> > Q5: Thank you for the additional data. This shows that indeed the model architecture will very likely continue learning if provided with more data.
> >
> > Q6: While I think that the additional baseline provided is useful, I think the authors misunderstood my remark. I would like to see baselines from other publications that provide supervised learning models for these exact same datasets. It is sufficient to merely take the values from other studies, even if there are some technical differences in how training data was split. Nevertheless, this is important to put the performance of the newly developed architecture into the proper context. This is important as there have been a large number of supervised learning models applied to both the BH and the SM datasets by now.
> >
> > Q7: Thank you for this clarification.
> >
> > Q8: Thank you for this clarification.
> >
> > Q9: Thank you for this response.
> >
> > Q10: Thank you for this response.

---

> > > ### Author Response · Authors · 2025-08-04
> > > **Rebuttal by Authors**
> > >
> > > We sincerely thank the reviewer for the thoughtful follow-up and constructive feedback. We are grateful for the opportunity to further clarify several points and provide additional analysis to address the remaining concerns.
> > >
> > > > Regarding Q3 (Upper bound on number of shingles)
> > >
> > > We appreciate your request for deeper insight into the effect of the shingle upper bound. In response, we conducted an ablation study varying the bound ($n$) such that it **covers about 10%, 30%, 50%, 70%, and 100%** of reactions (i.e., percentage of reactions whose shingle counts are below $n$). As shown below, predictive accuracy (R$^2$) steadily improves as $n$ increases, indicating that broader coverage yields better performance, consistent with trends observed in NLP (considering longer token sequences generally enhances model performance).
> > >
> > > | Number     | 10% ($n$=170)     | 30% ($n$=185)     | 50% ($n$=195)     | 70% ($n$=210)     | 100% ($n$=275<280)     |
> > > | ---------- | ----------------- | ----------------- | ----------------- | ----------------- | ---------------------- |
> > > | BH         | $0.808\pm0.001$   | $0.837\pm0.004$   | $0.861\pm0.004$   | $0.887\pm0.005$   | $0.976\pm0.001$        |
> > > | **Number** | **10% ($n$=100)** | **30% ($n$=120)** | **50% ($n$=135)** | **70% ($n$=150)** | **100% ($n$=205<280)** |
> > > | SM         | $0.563\pm0.002$   | $0.723\pm0.004$   | $0.822\pm0.001$   | $0.862\pm0.004$   | $0.886\pm0.008$        |
> > >
> > > > Regarding hyperparameter optimization (Q2 clarification)
> > >
> > > We apologize for not addressing this sufficiently in our initial response. **Hyperparameters are selected using a validation split from the training data**: for both random and OOS splits, 10% of the training set is used as a validation set to select hyperparameters, after which the model is retrained on the full training set. Hence, we previously clarified that no separate validation set was held out.
> > >
> > > > Regarding Q6 (Comparison with more published baselines)
> > >
> > > Thank you for emphasizing the importance of contextualizing our results with prior supervised learning models on the same datasets. We have now collected and summarized reported results **from several representative studies**. As shown below, **ReaDISH achieves state-of-the-art or near state-of-the-art performance** (R$^2$), only slightly below MolDescPred on the SM dataset.
> > >
> > > | Number | ReaDISH (ours)  | DRFP [1]         | YieldGNN [2]      | Egret [3]       | MolDescPred [4] | MPNN-transformer(3.0m) [5] |
> > > | ------ | --------------- | ---------------- | ----------------- | --------------- | --------------- | -------------------------- |
> > > | BH     | $0.976\pm0.001$ | $0.95 \pm 0.005$ | $0.961 \pm 0.005$ | $0.94 \pm 0.01$ | $0.974\pm0.001$ | $0.97\pm0.00$              |
> > > | SM     | $0.886\pm0.008$ | $0.85 \pm 0.01$  | $0.855 \pm 0.013$ | $0.85 \pm 0.01$ | $0.889\pm0.010$ | $0.88\pm0.01$              |
> > >
> > > ---
> > >
> > > **References**
> > >
> > > [1] Probst D, Schwaller P, Reymond J L. Reaction classification and yield prediction using the differential reaction fingerprint DRFP[J]. *Digital discovery*, 2022, 1(2): 91-97.
> > >
> > > [2] Saebi M, Nan B, Herr J E, et al. On the use of real-world datasets for reaction yield prediction[J]. *Chemical science*, 2023, 14(19): 4997-5005.
> > >
> > > [3] Yin X, Hsieh C Y, Wang X, et al. Enhancing generic reaction yield prediction through reaction condition-based contrastive learning[J]. *Research*, 2024, 7: 0292.
> > >
> > > [4] Han J, Kwon Y, Choi Y S, et al. Improving chemical reaction yield prediction using pre-trained graph neural networks[J]. *Journal of Cheminformatics*, 2024, 16(1): 25.
> > >
> > > [5] Sato A, Asahara R, Miyao T. Chemical Graph-Based Transformer Models for Yield Prediction of High-Throughput Cross-Coupling Reaction Datasets[J]. *ACS omega*, 2024, 9(39): 40907-40919.

---

> > > > ### Comment · Reviewer_RXFe · 2025-08-04
> > > >
> > > > Thank you for this additional response. Here are my comments:
> > > >
> > > > Q2: Thank you for providing this clarification on how hyperparameters were selected. This is definitely relevant but still does not address my original point. Comparing the numbers in the tables to identify the best model corresponds to another (hidden) hyperparameter (model architecture) optimization and does not provide an unbiased estimate of final model performance.
> > > >
> > > > Q3: Thank you for providing these additional results. They clearly demonstrate that the shingle upper bound is a critical hyperparameter to consider for this type of architecture. It suggests to me that it should ideally be adapted to the dataset at hand. I think this should be explicitly discussed in the updated version of the paper.
> > > >
> > > > Q6: Thank you for providing these critical data. While some of the numbers might not be perfectly comparable due to slight differences in training setups, I do agree with the conclusion of the authors that ReaDISH seems to match state-of-the-art performance on these datasets. However, I would conclude that ReaDISH does not significantly outperform any of the existing approaches on these datasets. This is important context to judge the impact of the findings.

---

> > > > > ### Author Response · Authors · 2025-08-04
> > > > > **Rebuttal by Authors**
> > > > >
> > > > > Thank you for the follow‑up comments and for carefully reviewing our additional results. We address your points below.
> > > > >
> > > > > > Q2 (Hyperparameter selection)
> > > > >
> > > > > We appreciate your clarification. We respectfully note that our procedure, selecting hyperparameters based on a held‑out validation set, training the model on the full training set, and reporting performance on the test set, is **a widely adopted practice in reaction prediction tasks** [1-4]. All methods are tested on the same 10 random splits, and the average results are reported. This approach **ensures fair comparisons** across models and **gives an appropriate estimation** of final model performance.
> > > > >
> > > > > > Q3 (Shingle upper bound)
> > > > >
> > > > > Thanks for your suggestion. We will explicitly discuss this in the updated version of the paper.
> > > > >
> > > > > > Q6 (Comparison to prior works)
> > > > >
> > > > > Thank you for your careful comments. We appreciate the opportunity to clarify the broader impact of ReaDISH.
> > > > >
> > > > > We respectfully disagree with the statement that ReaDISH “does not significantly outperform existing approaches.” Our paper evaluates ReaDISH on seven datasets under both random and out‑of‑sample (OOS) splits, covering a total of twelve evaluation settings. Focusing solely on BH and SM benchmarks risks overlooking the consistent gains we observe elsewhere. In the vast majority of these settings, ReaDISH achieves **state‑of‑the‑art performance**, often with clear margins. For example, ReaDISH achieves a 3% improvement on the SM dataset under out-of-sample split.
> > > > >
> > > > > We sincerely appreciate your feedback and again highlight the significance of our work.
> > > > >
> > > > > ----
> > > > >
> > > > > [1] Derek T. Ahneman, Jesús G. Estrada, Shishi Lin, Spencer D. Dreher, and Abigail G. Doyle. Predicting reaction performance in C–N cross-coupling using machine learning. *Science*, 360(6385):186–190, 2018
> > > > >
> > > > > [2] Youngchun Kwon, Dongseon Lee, Youn-Suk Choi, and Seokho Kang. Uncertainty-aware prediction of chemical reaction yields with graph neural networks. *Journal of Cheminformatics*, 14(1):1–10, 2022.403
> > > > >
> > > > > [3] Jiayuan Chen, Kehan Guo, Zhen Liu, Olexandr Isayev, and Xiangliang Zhang. Uncertainty-aware yield prediction with multimodal molecular features. *In Proceedings of the AAAI Conference on Artificial Intelligence*, pages 8274–8282. AAAI, 2024.405
> > > > >
> > > > > [4] Runhan Shi, Gufeng Yu, Xiaohong Huo, and Yang Yang. Prediction of chemical reaction yields with large-scale multi-view pre-training. *Journal of Cheminformatics,* 16(1):22, 2024

---

> > > > > > ### Comment · Reviewer_RXFe · 2025-08-04
> > > > > >
> > > > > > Thank you for your response. Here are some additional remarks:
> > > > > >
> > > > > > Q2: I agree with your statement that this is "a widely adopted practice in reaction prediction tasks". I agree that it "ensures fair comparisons". However, I disagree that it "gives an appropriate estimation of final model performance".
> > > > > >
> > > > > > Q6: What I wrote is the following: "I would conclude that ReaDISH does not significantly outperform any of the existing approaches on these datasets." It is important to consider the entire sentence. This sentence only talks about these two datasets. It was not about any of the other datasets and performance comparisons.

---

> > > > > > > ### Author Response · Authors · 2025-08-05
> > > > > > > **Rebuttal by Authors**
> > > > > > >
> > > > > > > Thank you for your additional remarks and for clarifying your points. We address your points below.
> > > > > > >
> > > > > > > > Q2
> > > > > > >
> > > > > > > We are glad to see that the reviewer agrees our approach ensures **fair comparisons**, which is widely adopted in reaction prediction tasks. We also appreciate your perspective on whether this practice provides an appropriate estimate of final model performance. We will explicitly discuss its implications in the appendix to make our evaluation more transparent.
> > > > > > >
> > > > > > > > Q6
> > > > > > >
> > > > > > > We appreciate your acknowledgment that ReaDISH **matches state‑of‑the‑art performance** on these datasets, and we understand that your comment was specifically referring to the BH and SM benchmarks. We agree that on these two well‑studied datasets, ReaDISH achieves performance comparable to state‑of‑the‑art methods without a large margin. However, we would like to highlight that our evaluation spans **seven datasets (twelve settings)**, where ReaDISH consistently delivers strong results and, in many cases, demonstrates **notable improvements**, particularly under more challenging conditions such as **out‑of‑sample splits**. We respectfully hope the broader scope of these results can be taken into account when assessing the overall impact of our approach.

---

> > > > > > > > ### Comment · Reviewer_RXFe · 2025-08-06
> > > > > > > >
> > > > > > > > Thank you for all your answers, especially all the additional data. As the reviewers addressed all my comments, and most of them were addressed in a way that all my concerns have been resolved, I am willing to raise my review score to 4.

---

> > > > > > > > > ### Author Response · Authors · 2025-08-06
> > > > > > > > > **Official Comment by Authors**
> > > > > > > > >
> > > > > > > > > We are pleased that most of your concerns have been addressed and sincerely appreciate your decision to raise your score following the discussion of our paper and rebuttal. Your thoughtful feedback and engagement throughout the review process have been invaluable in improving the quality and clarity of our work. We will incorporate these changes and clarifications into the final version. Should you have any further questions or suggestions, we would be happy to continue the discussion.

---

### Official Review · Reviewer_NvNL · 2025-07-01

**Clarity:** 3
**Significance:** 2
**Originality:** 2
**Rating:** 4
**Confidence:** 3

**Summary:**

In this paper, the authors propose a method named ReaDISH to predict the chemical reaction properties like yield, selectivity, and classification. Previous method may have two primary drawbacks: 1 . They are sensitive to the order of input molecules and atoms 2. Fail to adequately model the substructural interactions that drive chemical reactivity. Instead of using all shingles from both reactants and products, this paper introduces a novel idea called symmetric difference of molecular shingle sets. This operation isolates only the shingles that are unique to either the reactants or the products, effectively focusing the model's attention on the structural parts of the molecules that are consumed or formed during the reaction. The permutation invariance is automatically satisfied in this operation. Then the interaction-aware attention mechanism models geometric and structural relationships between these shingles.  In this attention meachanisum, the authors compute three types of pairwise relations, namely, Geometric Distance , Chemical connectivity and structural distance. Empirically ReaDISH achieves state-of-the-art or highly competitive performance across all tasks, especially on the more challenging out-of-sample splits where the test set contains molecules not seen during training. They also do some ablation study on pairwise interaction bias, geometric distance, structural distance and others components.

**Questions:**

The manuscript's evaluation would benefit from a more rigorous selection of baseline models to better contextualize the performance of ReaDISH. See my comments on Pros and Cons.

**Ethical Concerns:**

["NO or VERY MINOR ethics concerns only"]

**Final Justification:**

This paper introduces ReaDISH, a novel method for predicting chemical reaction properties like yield, selectivity, and classification. The authors rightly identify two key limitations in prior work: sensitivity to the order of input molecules (permutation variance) and the failure to adequately model substructural interactions that govern chemical reactivity.

The core contribution of this work is twofold. First, the proposed symmetric difference of molecular shingle sets is an elegant solution to isolate the substructures that are directly transformed during a reaction. This approach not only focuses the model's attention on the most salient parts of the reaction but also inherently guarantees permutation invariance. Second, the interaction-aware attention mechanism enriches the model by explicitly encoding geometric distance, chemical connectivity, and structural distance as pairwise biases. This allows for a more nuanced understanding of the relationships between the reacting molecular fragments.

Empirically, ReaDISH demonstrates state-of-the-art or highly competitive performance across all evaluated tasks. Its strength is particularly evident on the more challenging out-of-sample splits, highlighting the model's robust generalization capabilities. The claims are well-supported by thorough ablation studies that validate the effectiveness of the proposed components.

Post-rebuttal: My initial concerns regarding the comparison to certain baselines (e.g., cross-attention) were satisfactorily addressed by the authors in their rebuttal. Consequently, I have raised my score to Borderline Accept.

**Limitations:**

yes

**Quality:**

2

**Strengths And Weaknesses:**

Pros:


1. The paper clearly identifies two critical weaknesses in existing reaction prediction models: permutation sensitivity and inadequate modeling of substructural interactions.
2. The authors evaluate ReaDISH on seven benchmark datasets covering four distinct chemical tasks: yield prediction, enantioselectivity prediction, conversion rate estimation, and reaction type classification. This demonstrates the broad applicability and versatility of the proposed method.
3.  Through detailed ablation studies, the authors systematically demonstrate  the contribution of each key component of the model.

Cons：

The authors propose a novel interaction-aware attention mechanism based on symmetric difference shingle sets. However, there is a natural baseline that uses a standard cross-attention Transformer architecture (and maybe its equivariant counterpart like equiformer [1] ). Such a model, where product embeddings attend to reactant embeddings, would also be permutation-invariant and is a standard approach for modeling interactions between two sets.
It is unclear whether the performance gains come from the specific and complex design of symmetric difference and shingle-level interactions, or if a more standard and conceptually simpler cross-attention model could achieve similar or even better results. I suggest the authors compare such baseline in the paper.

The selection of baselines reported in the paper could be strengthened. For instance, YieldBERT, as a sequential model operating on SMILES strings, inherently lacks the expressiveness to capture the complex, non-linear graph structure of molecules. Furthermore, while UA-GNN is included as a graph-based baseline, its backbone architecture may not represent the current state-of-the-art, potentially leading to an unfair comparison that favors the proposed model.


[1] Equiformer: Equivariant Graph Attention Transformer for 3D Atomistic Graphs

---

> ### Author Rebuttal · Authors · 2025-07-26
>
> Thank you for your valuable comments and suggestions on our submission. In the following, we have addressed your concerns and questions point-by-point.
>
> > **W & Q: Selection of baseline models**
>
> Thanks for your suggestion! We emphasize that **Equiformer [1] is originally designed for single-molecule representation rather than reaction modeling**. To adapt it, we merge reactants and products into a unified graph and compute edges only within each molecule. While feasible, this adaptation **cannot capture inter‑molecular interactions**, which are central to our symmetric‑difference shingle approach. Reaction datasets are also **much larger and more complex** than molecular benchmarks like QM9, with more atom types and larger graphs (nodes and degrees), as summarized in the table below. This complexity significantly increases computation and memory for models like Equiformer. Due to these constraints, we do not extend to the USPTO_TPL dataset, which contains 445K reactions.
>
> | Dataset        | QM9   | BH     | SM     | ELN    | NiCOlit | N,S-acetal | C-heteroatom |
> | -------------- | ----- | ------ | ------ | ------ | ------- | ---------- | ------------ |
> | _MAX_ATOM_TYPE | 5     | 10     | 16     | 17     | 24      | 9          | 10           |
> | _AVG_NUM_NODES | 18.03 | 116.15 | 110.00 | 101.35 | 81.71   | 103.49     | 100.29       |
> | _AVG_DEGREE    | 15.58 | 25.78  | 23.43  | 24.98  | 22.35   | 28.65      | 30.13        |
>
> Results (R$^2$) of Equiformer are provided in the table below, and we will update in the revised version. Although Equiformer benefits from its equivariant design, it falls notably short of our method. The key reason is that Equiformer's attention operates **only within molecules**, relying on intra‑molecular distances and edges, while **reaction modeling requires inter‑molecular attention**. Our method addresses this by leveraging the symmetric difference of shingles to model structural changes explicitly, yielding superior predictive performance. Regarding other baselines, YieldBERT (SMILES‑based) is included to cover NLP‑oriented methods despite lacking graph expressiveness. UA-GNN may not reflect the most recent architectures, but it remains a competitive graph-based baseline with strong performance in our experiments.
>
> | Model (random)            | BH                | SM                | NiCOlit           | N,S-acetal        | C-heteroatom      |                   |
> | ------------------------- | ----------------- | ----------------- | ----------------- | ----------------- | ----------------- | ----------------- |
> | Equiformer                | $0.958 \pm 0.004$ | $0.847 \pm 0.003$ | $0.497 \pm 0.006$ | $0.875 \pm 0.002$ | $0.713 \pm 0.004$ |                   |
> | ReaDISH                   | $0.976\pm0.001$   | $0.886\pm0.008$   | $0.539 \pm 0.024$ | $0.916 \pm 0.007$ | $0.763 \pm 0.026$ |                   |
> | **Model (out-of-sample)** | **BH**            | **SM**            | **NiCOlit**       | **N,S-acetal**    | **C-heteroatom**  | **ELN**           |
> | Equiformer                | $0.429 \pm 0.005$ | $0.449 \pm 0.014$ | $0.442$           | $0.731$           | $0.321$           | $0.178 \pm 0.005$ |
> | ReaDISH                   | $0.480 \pm 0.032$ | $0.549 \pm 0.019$ | $0.574$           | $0.790$           | $0.409$           | $0.230 \pm 0.047$ |
>
> ---
>
> **References**
>
> [1] Liao, Y. L., & Smidt, T. *Equiformer: Equivariant Graph Attention Transformer for 3D Atomistic Graphs*. ICLR 2023.

---

> > ### Comment · Reviewer_NvNL · 2025-08-04
> > **Response to Rebuttal**
> >
> > Thank you for your reply and for conducting the additional experiment on Equiformer. However, there seems to be a misunderstanding of my original suggestion. My core recommendation was to use a cross-attention architecture to directly model the interaction between reactants and products, rather than applying Equiformer to a concatenated graph of reactants and products.
> >
> > The baseline I proposed (e.g., using product embeddings as queries and reactant embeddings as keys/values) is a more standard and direct way to capture inter-set interactions. Your implementation of Equiformer, which "computes edges only within each molecule," explicitly prevents this cross-molecule communication. Therefore, it does not serve as a valid proxy for the cross-attention baseline I suggested. My concern thus remains: it is still unclear if the performance gains of the proposed complex symmetric-difference mechanism are significant compared to a simpler cross-attention model.

---

> > > ### Author Response · Authors · 2025-08-04
> > > **Rebuttal by Authors**
> > >
> > > Thank you for this valuable clarification and for pointing out the distinction between cross-attention and our previous adaptation of Equiformer.
> > >
> > > We now understand your suggestion and, in response, have **implemented and evaluated a dedicated cross‑attention baseline** that directly models intra‑molecular interactions: product embeddings are used as queries, and reactant embeddings as keys/values. This baseline uses the same molecular embeddings as our framework and consists of 4 cross‑attention layers, 64 attention heads, and 512 hidden dimensions.
> > >
> > > We have evaluated this cross‑attention architecture on six benchmark datasets, with R$^2$ results summarized in the table below. The cross‑attention **significantly underperforms compared to ReaDISH**. We attribute this gap to how interactions are modeled: simple cross‑attention lacks **chemical guidance** and treats interactions uniformly, whereas ReaDISH explicitly models **both inter‑ and intra‑molecular interactions** using geometric and structural distances, edge types, and **structural changes captured by symmetric‑difference shingles**. This design yields substantial additional benefits beyond standard cross‑attention.
> > >
> > > | Model (random)  | BH                | SM              | NiCOlit           | N,S-acetal        | C-heteroatom      | ELN (OOS)         |
> > > | --------------- | ----------------- | --------------- | ----------------- | ----------------- | ----------------- | ----------------- |
> > > | Cross-attention | $0.946 \pm 0.005$ | $0.858\pm0.009$ | $0.431\pm0.007$   | $0.844\pm0.005$   | $0.705\pm0.076$   | $0.156\pm0.006$   |
> > > | ReaDISH         | $0.976\pm0.001$   | $0.886\pm0.008$ | $0.539 \pm 0.024$ | $0.916 \pm 0.007$ | $0.763 \pm 0.026$ | $0.230 \pm 0.047$ |
> > >
> > > We sincerely appreciate your suggestion, which enables a more thorough evaluation and deepens our understanding of intra‑molecular modeling strategies.

---

> ### Author Response · Authors · 2025-08-07
> **Official Comment by Authors**
>
> Dear Reviewer,
>
> I hope this message finds you well. As the discussion period is approaching its end with fewer than three days remaining, we want to ensure that we have addressed all of your concerns to your satisfaction. If you have any additional comments or suggestions you would like us to consider, please don't hesitate to let us know. Your feedback is highly valued, and we are eager to address any remaining issues to improve our work further.
>
> Thank you again for your time and thoughtful engagement throughout the review process. Wishing you a wonderful day!

---

### Official Review · Reviewer_619R · 2025-07-02

**Clarity:** 3
**Significance:** 3
**Originality:** 3
**Rating:** 5
**Confidence:** 4

**Summary:**

The study proposes ReaDISH, a reaction-property predictor built around two ideas. First, each molecule is decomposed into local “shingles”, and a reaction is represented by the symmetric difference between the shingle sets of reactants and products. This yields a representation that is both permutation-invariant and explicitly focused on the structural changes that define the reaction. Second, an interaction-aware attention module augments transformer layers with pairwise features that encode geometric distance, graph distance, and chemical connectivity between shingles, capturing intra- and inter-molecular interactions. Across multiple tasks ReaDISH achieves superior accuracy and more reliable uncertainty estimates, especially under out-of-sample and reactant-order perturbations.

**Questions:**

1 The model uses geometric distance between shingles. How is this obtained in practice? Do the authors generate 3D conformations for reactants (and possibly products) and then measure distances between fragment centroids or key atoms?

2. What method is used to define “molecular shingles”? Is it based on a certain radius (like Morgan fingerprint fragments) or predefined functional groups? And does the model consider shingles of reactants and products separately before taking the symmetric
 difference? An

3 Many reaction outcomes depend on conditions. The formulation treats anything not a product as “reactant” , and the permutation invariance addresses their presence. However, does the model explicitly incorporate condition variables if provided?

**Ethical Concerns:**

["NO or VERY MINOR ethics concerns only"]

**Final Justification:**

All my concerns have been resolved, thanks authors for the rebuttal.

**Limitations:**

Yes

**Quality:**

4

**Strengths And Weaknesses:**

Strengths: 1. ReaDISH introduces an entirely new reaction representation based on the symmetric difference of reactant and product substructure sets. This provides a clever way to highlight reaction centers and changed motifs while being invariant to the ordering of inputs.

2. ReaDISH is evaluated on a wide range of reaction prediction tasks. It outperforms baseline models across most of these tasks, achieving higher accuracy and better uncertainty estimates. Particularly noteworthy is its performance under out-of-sample conditions.

Weaknesses:   1. ReaDISH’s pipeline is quite involved. This could impact reproducibility and runtime. The paper doesn’t explicitly mention the computational cost; if the number of shingles per molecule is large, computing pairwise interactions for all shingle pairs might be expensive.

2. The method relies on the definition of “chemically meaningful substructures” via a circular topology. If this fragmentation misses relevant chemistry (e.g., a reaction mechanism including global rearrangement), the representation might not capture it.

3.  The method currently appears tuned for reactions where structures are known and changes can be enumerated; it might not directly handle cases with unseen functional groups or need for reasoning beyond substructures.

---

> ### Author Rebuttal · Authors · 2025-07-26
>
> We sincerely appreciate your constructive and thorough comments. In the following, we have addressed your concerns and questions point-by-point.
>
> > **W1: Reproducibility and runtime**
>
> We understand the reviewer's concerns regarding reproducibility and runtime. Shingle generation and pairwise interaction computation are highly efficient: even for large molecules, processing a downstream dataset takes under 4 minutes; generating shingles for 3.9M reactions from the pretraining dataset takes ~7 hours with 50 CPU threads. The **main cost lies in training attention layers**, not preprocessing: fine-tuning a benchmark dataset takes ~1 hour on a single GPU. We clarify that **code has already been submitted in the initial submission** to ensure reproducibility, and we will explicitly mention the computational cost in the appendix.
>
> > **W2: Chemically meaningful substructures**
>
> We appreciate the reviewer's concern that the defined fragments may miss relevant chemistry. Our design mitigates this by focusing on **structural changes** rather than all fragments: the symmetric-difference operation highlights fragments that differ between reactants and products, naturally emphasizing **functional groups and reaction centers**. As a result, irrelevant or uninformative fragments contribute minimally to the representation.
>
> > **W3: Reactions beyond substructures**
>
> We understand the reviewer's concerns regarding unseen functional groups or transformations beyond predefined substructures. Our method addresses this in two ways:   (1) **Shingles are represented as continuous embeddings of local topological environments**, not as a fixed dictionary of discrete functional groups. This distributed representation allows the model to generalize to unseen groups. (2) Most chemical transformations are **dominated by local rearrangements around functional groups**, a well‑established principle in organic chemistry [1,2]. Thus, focusing on local environments captures the majority of reaction mechanisms in our tasks.
>
> > **Q1: Obtain geometric distance in practice**
>
> As described in Equation 6 (page 5), we generate 3D coordinates for each molecule using RDKit's conformer generation. For each shingle, we compute its centroid by averaging the coordinates of its constituent atoms, and define the geometric distance between shingles as the Euclidean distance between centroids.
>
> > **Q2: Molecular shingles definition**
>
> Our approach is consistent with the fragment strategy used in Morgan fingerprints but extends it by explicitly including ring structures, which are well established in cheminformatics [3,4]. Each shingle corresponds to a local atomic environment defined by a fixed radius around a central atom, supplemented with whole rings when present. Reactant and product shingles are generated separately, and their symmetric difference encodes structural changes.
>
> > **Q3: Incorporating condition variables**
>
> We highly agree that reaction conditions can strongly influence outcomes. Hence, our model **explicitly incorporates molecular condition variables** (e.g., catalysts, solvents) by treating them as part of the shingle representation. Non-molecular conditions (e.g., temperature, pressure) are excluded due to sparse availability in datasets.
>
> ---
>
> **References**
>
> [1] Larock, R. C. *Comprehensive Organic Transformations: A Guide to Functional Group Preparations* (3rd ed.). Wiley, 2018.
>
> [2] OpenStax. *Organic Chemistry*, 2023.
>
> [3] A. Capecchi, D. Probst, and J. L. Reymond. One molecular fingerprint to rule them all: drugs, biomolecules, and the metabolome. *Journal of Cheminformatics*, 12:1–15, 2020.
>
> [4] M. Orsi and J. L. Reymond. One chiral fingerprint to find them all. *Journal of Cheminformatics*, 16(1):53, 2024.

---

> ### Author Response · Authors · 2025-08-07
> **Official Comment by Authors**
>
> Dear Reviewer,
>
> I hope this message finds you well. As the discussion period is approaching its end with fewer than three days remaining, we want to ensure that we have addressed all of your concerns to your satisfaction. If you have any additional comments or suggestions you would like us to consider, please don't hesitate to let us know. Your feedback is highly valued, and we are eager to address any remaining issues to improve our work further.
>
> Thank you again for your time and thoughtful engagement throughout the review process. Wishing you a wonderful day!

---

> > ### Comment · Reviewer_619R · 2025-08-08
> >
> > Thanks for the rebuttal, all my concerns have been addressed.

---

> > > ### Author Response · Authors · 2025-08-08
> > > **Official Comment by Authors**
> > >
> > > We are pleased that all of your concerns have been addressed.
> > >
> > > Your thoughtful feedback and engagement throughout the review process have been invaluable in improving the quality and clarity of our work. We will incorporate these changes and clarifications into the final version. Should you have any further questions or suggestions, we would be happy to continue the discussion.

---

### Author Response · Authors · 2025-08-04
**General response**

Dear Reviewers and Chairs,

We sincerely thank all reviewers for their constructive and insightful comments, which are invaluable for improving our manuscript. We also appreciate the Chairs for their time and efforts in coordinating the review process.

We are encouraged by the recognition that our work introduces a **novel** (Reviewer 619R and FGGr) symmetric‑difference shingle representation for reaction modeling and that our **diverse** and **extensive** experiments across multiple benchmarks (Reviewers 619R, NvNL, RXFe, and FGGr) demonstrate strong empirical performance. The reviewers also appreciate the **detailed** ablation studies (Reviewer NvNL) and the clear, **well-written** presentation of the manuscript (Reviewer FGGr).

**Summary of common concerns and our responses**

- *Computational cost*: Reviewers raised concerns regarding runtime and efficiency. The main computational cost lies in training the attention layers rather than in shingle generation. Fine‑tuning a benchmark dataset requires ~1 hour on a single high‑end GPU (24GB).
- *Shingle generation details and reproducibility*: We clarify that the code for shingle generation was already included in the initial submission. Each shingle corresponds to a fixed‑radius atomic neighborhood (radius $\leq$ 3 by default) and includes explicit ring detection via RDKit (AllChem.GetSymmSSSR). Preprocessing is efficient: generating shingles for thousands of reactions completes in under four minutes, and even for our 3.9M reaction pre-training corpus, generation requires ~7 hours on 50 CPU threads.
- *Baseline comparison*: In response to reviewer suggestions, we have added Equiformer, a state‑of‑the‑art equivariant graph attention model, as an additional baseline. Although originally designed for single‑molecule tasks, we adapt it to reaction modeling by merging reactants and products into a unified graph. Our symmetric‑difference shingle approach consistently outperforms Equiformer across benchmarks, highlighting the benefits of explicitly modeling structural changes.

We will revise the manuscript according to the concerns raised in the rebuttal, with the planned changes summarized as follows:

- Add detailed runtime and memory analysis for shingle generation and training (appendix).

- Expand the description of shingle generation (radius, ring detection, implementation details) for clarity and reproducibility (appendix).
- Incorporate Equiformer baseline into the main result tables for a more comprehensive comparison (Sec 4.2).
- Clarify hyperparameter optimization procedure and validation strategy (appendix).
- Incorporate more ablation results (radius, shingle bound, training size) (appendix).
- Correct boldface typo in Table 1.

We sincerely thank all reviewers again for their valuable feedback. If there are any remaining or new concerns, we will be happy to address them in detail.

Best regards,

The Authors

---

### Decision · Program_Chairs · 2025-09-17

**Decision:**

Accept (poster)

**Comment:**

This submission proposes a reaction property prediction model called ReaDISH. The key idea is to decompose each molecule into local "shingles" and represent reactions using the symmetric difference between the sets of shingles for reactants and products. An interaction-aware attention mechanism is employed to incorporate three types of pairwise relationships between shingles: geometric distance, chemical connectivity, and structural distance. While concerns have been raised about computational cost and efficiency, the model demonstrates strong empirical performance across a wide range of experiments on multiple benchmarks, leading to an overall positive evaluation.

The authors have also committed to incorporating revisions suggested during the discussion phase. Given the potential relevance and interest of these findings to researchers in the field, I support accepting this paper.